# Differences in interactions between transmembrane domains tune the activation of metabotropic glutamate receptors

Jordana K Thibado[1], Jean-Yves Tano[2], Joon Lee[3], Leslie Salas-Estrada[4], Davide Provasi[4], Alexa Strauss[5], Joao Marcelo Lamim Ribeiro[4], Guoqing Xiang[3], Johannes Broichhagen[6], Marta Filizola[4], Martin J Lohse[2,7], Joshua Levitz[1,3,5]*

[1]Physiology, Biophysics and Systems Biology Graduate Program, Weill Cornell Graduate School of Medical Sciences, New York, United States; [2]Max Delbrück Center for Molecular Medicine, Berlin, Germany; [3]Department of Biochemistry, Weill Cornell Medicine, New York, United States; [4]Department of Pharmacological Sciences, Icahn School of Medicine at Mount Sinai, New York, United States; [5]Tri-Institutional PhD Program in Chemical Biology, New York, United States; [6]Leibniz-Forschungsinstitut für Molekulare Pharmakologie, Berlin, Germany; [7]ISAR Bioscience Institute, Planegg-Munich, Germany

*For correspondence:
jtl2003@med.cornell.edu

Competing interests: The authors declare that no competing interests exist.

**Abstract** The metabotropic glutamate receptors (mGluRs) form a family of neuromodulatory G-protein-coupled receptors that contain both a seven-helix transmembrane domain (TMD) and a large extracellular ligand-binding domain (LBD) which enables stable dimerization. Although numerous studies have revealed variability across subtypes in the initial activation steps at the level of LBD dimers, an understanding of inter-TMD interaction and rearrangement remains limited. Here, we use a combination of single molecule fluorescence, molecular dynamics, functional assays, and conformational sensors to reveal that distinct TMD assembly properties drive differences between mGluR subtypes. We uncover a variable region within transmembrane helix 4 (TM4) that contributes to homo- and heterodimerization in a subtype-specific manner and tunes orthosteric, allosteric, and basal activation. We also confirm a critical role for a conserved inter-TM6 interface in stabilizing the active state during orthosteric or allosteric activation. Together this study shows that inter-TMD assembly and dynamic rearrangement drive mGluR function with distinct properties between subtypes.

## Introduction

Assembly into multi-subunit complexes is an essential aspect of many membrane receptors. Dimerization or higher order oligomerization can shape the ligand sensitivity or signaling dynamics of a protein complex by producing various forms of allostery and cooperativity or mediate the formation of binding, functional, or regulatory sites (*Changeux and Christopoulos, 2016*; *Katzung, 2004*). In many cases, the formation of a multi-subunit complex is required for function. For example, ligand-gated ion channels require multiple subunits to form a functional pore, making assembly into complexes an absolute necessity. In the case of the vast superfamily of G-protein-coupled receptors (GPCRs) the role of quaternary structure has remained more enigmatic and is likely different for the various subfamilies and their individual members. The proposed dimerization of family A GPCRs, such as rhodopsin or the beta-adrenergic receptors, has sparked intense debate (*Ferré et al., 2014*; *Milligan et al., 2019*; *Sleno and Hébert, 2019*) with recent studies showing that dimerization is

likely transient (*Hern et al., 2010*; *Kasai et al., 2011*; *Meral et al., 2018*; *Işbilir et al., 2020*; *Möller et al., 2020*; *Felce et al., 2017*), and is not generally required for G protein activation (*Bayburt et al., 2007*; *Whorton et al., 2008*; *Whorton et al., 2007*; *Kuszak et al., 2009*). Importantly, a structural understanding of GPCR dimerization has remained elusive with a wide range of dimer interfaces proposed for different GPCRs and, more recently, the emergence of the concept of dynamic, 'rolling' inter-subunit interfaces which inter-change and are associated with different functional states (*Gurevich and Gurevich, 2008*; *Meral et al., 2018*; *Sleno and Hébert, 2019*; *Dijkman et al., 2018*; *Jin et al., 2018*).

One major exception to the secondary, modulatory role for dimerization in the function of GPCRs are the family C GPCRs, including the metabotropic glutamate receptors (mGluRs) and a number of other subfamilies (*Ellaithy et al., 2020*; *Pin and Bettler, 2016*). Family C GPCRs contain canonical seven-helix transmembrane domains (TMDs) but are distinguished by their large, bi-lobed, extracellular ligand binding domains (LBDs), which are essential for their constitutive homo- and heterodimerization (*Romano et al., 1996*; *Doumazane et al., 2011*; *Levitz et al., 2016*; *Lee et al., 2020*). Unlike family A GPCRs, reconstituted mGluR2 monomers are unable to undergo glutamate-mediated G-protein activation (*El Moustaine et al., 2012*), although they can be activated by TMD-targeting allosteric drugs. In response to orthosteric agonists, mGluR LBD dimers undergo dramatic rearrangement between conformations which both reshape and are controlled by a complex inter-LBD dimer interface, as deciphered through crystallographic (*Kunishima et al., 2000*; *Tsuchiya et al., 2002*; *Muto et al., 2007*; *Koehl et al., 2019*) and spectroscopic (*Levitz et al., 2016*; *Vafabakhsh et al., 2015*; *Olofsson et al., 2014*) studies. However, how such motions are transmitted to the TMDs, how TMDs can be activated directly by allosteric drugs and the role of associated inter-TMD interactions in this family of receptors remains unclear, although a variety of recent studies have provided insight.

A series of Förster resonance energy transfer (FRET) studies of full-length mGluR1 have shown that glutamate and other agonists produce relative motions between the TMDs that occur on fast millisecond time scales that precede eventual receptor activation (*Hlavackova et al., 2012*; *Marcaggi et al., 2009*; *Grushevskyi et al., 2019*; *Tateyama et al., 2004*; *Tateyama and Kubo, 2006*). Structural interpretation of such studies has been limited but the FRET efficiency increase seen with intersubunit sensors points to a rearrangement that increases inter-TMD interaction upon LBD activation. Notably, we recently developed an isolated TMD FRET sensor which showed that TMD-targeting allosteric drugs are able to produce intersubunit rearrangements autonomously without allosteric input from the LBDs (*Gutzeit et al., 2019*), pointing to a multi-state model of inter-TMD interaction and raising questions about TMD rearrangement in response to allosteric versus orthosteric activation. In line with these findings, a prior cross-linking study of full-length mGluR2 provided evidence for close proximity of TM4 and TM5 helices in inactive states and TM6 helices in active states (*Xue et al., 2015*). Despite such studies, the relative contribution of inter-TMD interactions within mGluR dimers remains unknown.

High-resolution structural studies have provided further insight but no clear consensus regarding TMD interfaces. Crystal structures of isolated mGluR1 (*Wu et al., 2014*) and mGluR5 (*Doré et al., 2014*; *Christopher et al., 2019*; *Christopher et al., 2015*) TMDs have shown TM1-mediated dimerization or monomers, respectively, while a cryogenic electron microscopy (cryo-EM) study of full-length mGluR5 (*Koehl et al., 2019*) showed no TMD interface in an inactive nanodisc-reconstituted form but a clear TM6 interface in a glutamate and positive allosteric modulator-bound pre-active state. Finally, a series of cryo-EM studies of the related GABA$_B$ receptors showed a lipid-mediated inactive TM5 interface that re-arranged to a TM6 interface in the presence of agonist and a positive allosteric modulator (*Mao and Shen, 2020*; *Papasergi-Scott et al., 2020*; *Park et al., 2020*; *Shaye et al., 2020*). Together these studies point to a role for TMD rearrangement in mGluR activation, but the steps prior to formation of a putative active TM6 interface are not clear.

Recent work has also revealed conformational heterogeneity between mGluR subtypes at the level of the LBDs that shape the activation properties of different homo- and heterodimers (*Levitz et al., 2016*; *Vafabakhsh et al., 2015*; *Habrian et al., 2019*). Such findings are in line with the notion that the eight mGluR subtypes are fine-tuned for their distinct roles within the synapse that require unique glutamate sensitivity, activation kinetics and other signaling properties (*Reiner and Levitz, 2018*). This raises the possibility that such variability between subtypes also exists at the TMDs and can be used as a lens to dissect the TMD activation mechanism. Intriguingly,

we recently used a single-molecule imaging approach to determine that while all isolated mGluR TMDs tested show a clear propensity for dimerization, the mGluR2 subtype shows substantially higher dimerization propensity than other subtypes (*Gutzeit et al., 2019*), providing a jumping off point for this study.

Here, we use a combination of optical, functional, and computational techniques to dissect the role of inter-TMD interaction in mGluR dimerization and activation with a focus on the highly homologous group II mGluRs (mGluR2 and mGluR3). Single-molecule fluorescence subunit counting experiments and coarse-grained (CG) molecular dynamics (MD) simulations reveal TMD-mediated differences in dimer assembly between mGluR2 and mGluR3. Using observations from sequence analysis, mutagenesis, and subunit counting, we isolate this effect to specific residues at the cytoplasmic end of transmembrane helix 4 (TM4). Functional experiments demonstrate that TM4 residues play a modulatory role in both orthosteric and allosteric activation of group II mGluRs and controlling basal activity. Using new inter-TMD FRET sensors in full-length mGluRs, we characterize how orthosteric drugs, allosteric modulators and TM4 residues influence the relative TMD conformational dynamics of group II mGluRs. Further, we demonstrate that activation associated inter-TMD dynamics involve reorientation away from a TM4 inactive interface toward a TM6 active interface by employing single molecule subunit counting experiments in the presence of ligands. Finally, we find that inter-TMD dimerization shows an intermediate level in mGluR2/3 heterodimers compared to homodimers, supporting the role of TMD interactions as mediators of the molecular diversity of mGluRs.

## Results

### The transmembrane domain mediates differences in dimerization between metabotropic glutamate receptors

Motivated by our finding that isolated mGluR TMDs dimerize to variable levels in the absence of extracellular domains (*Gutzeit et al., 2019*), we sought to further understand the molecular basis of mGluR TMD dimerization using single molecule pulldown (SiMPull). In this technique, detergent-solubilized receptor complexes are isolated from fresh cell lysate and sparsely immobilized via antibodies on a glass coverslip. Protein complexes are imaged using TIRF microscopy and photobleaching step analysis of complexes reports on receptor stoichiometry (*Jain et al., 2011*). We restrict our analysis to surface receptors by using N-terminally SNAP-tagged receptor constructs labeled with membrane-impermeable fluorophores. Expression and labeling of constructs in HEK 293 T cells with benzylguanine-conjugated fluorophore LD555 ('BG-LD555'; see Materials and methods) showed fluorescence labeling at the plasma membrane with minimal fluorescence inside the cell (*Figure 1A*). Receptors were immobilized using an HA-tag at the N-terminus of each construct that directly precedes the SNAP tag (*Figure 1B*). Consistent with previous work, SNAP-mGluR2-TMD molecules photobleached primarily in one-step (54.1%) or two-step (41.4%) events with a small population of ≥3 steps (4.5%) (*Figure 1C*). This represents a population of 83.1 ± 1.5% dimers when normalized based on the obligatory dimerization of SNAP-mGluR2 (*Figure 1D*; see Materials and methods) (*Gutzeit et al., 2019*). Importantly, we and others have shown that isolated mGluR TMD constructs maintain functionality based on downstream responses to allosteric modulators (*Goudet et al., 2004*; *El Moustaine et al., 2012*; *Koehl et al., 2019*; *Gutzeit et al., 2019*).

Since mGluR TMDs do not form strict dimers, we asked if our preparation maintains the cellular composition of monomers and dimers or if TMDs exchange with one another such that new dimers are formed. To test this, we either co-expressed HA-SNAP-mGluR2-TMD and CLIP-mGluR2-TMD and labeled the same coverslip with both BG-LD655 (for SNAP) and BC-DY547 (for CLIP) or transfected and labeled separate coverslips with either construct/fluorophore combination prior to mixing for 45 min during co-lysis (*Figure 1—figure supplement 1A–C*). Following lysis, protein complexes were isolated via an anti-HA antibody and all data was taken within 2 hr. As expected, in the positive co-expression control HA-SNAP-mGluR2-TMD was able to co-precipitate many spots for CLIP-mGluR2-TMD (*Figure 1—figure supplement 1D*). In contrast, we observed background levels of spots for the test condition where separate populations of cells were mixed following labeling (*Figure 1—figure supplement 1E,F*). This suggests that there is no exchange between TMD dimers within detergent on the timescale of our experiments and indicates that our technique can be

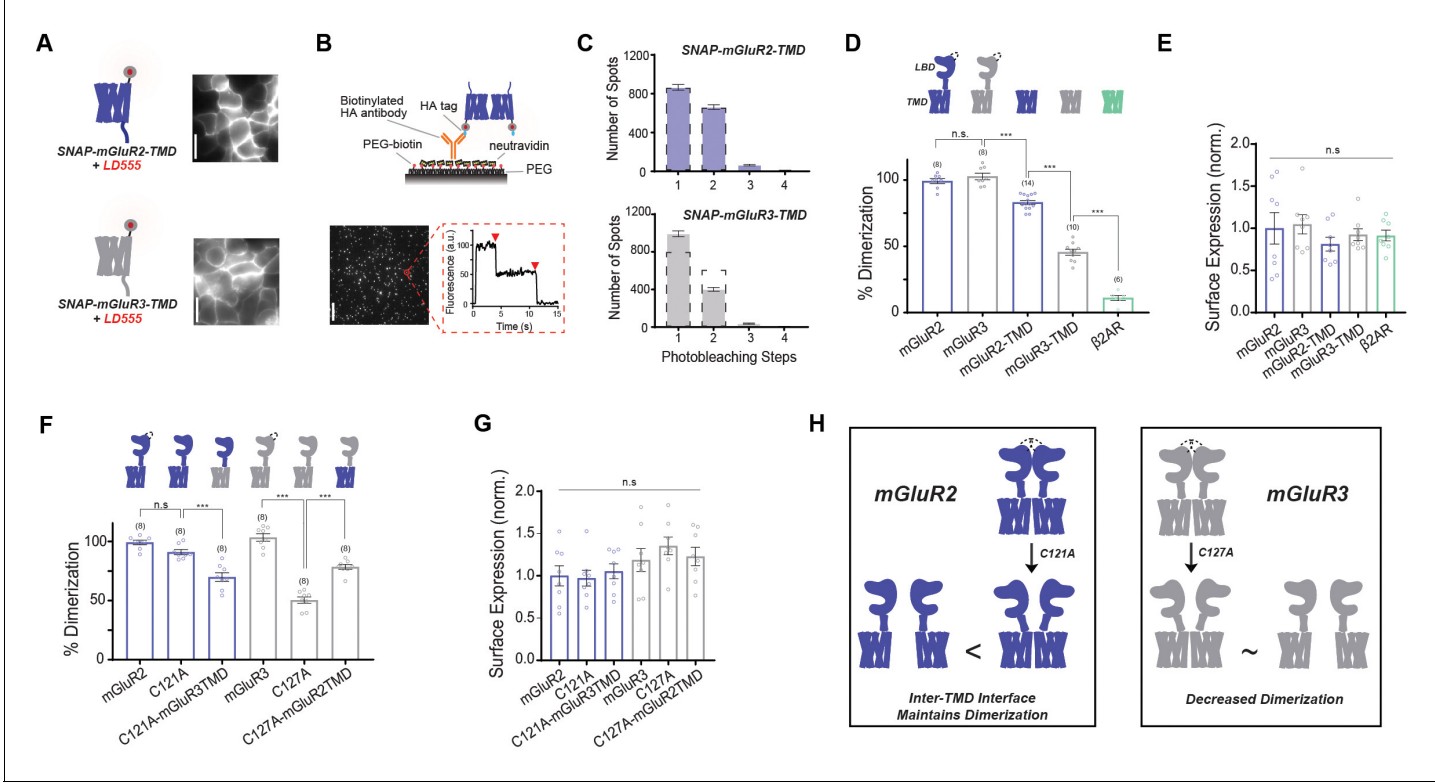

**Figure 1.** mGluR2 and mGluR3 transmembrane domains show different dimerization propensities in a single molecule pulldown assay. (**A**) Left, cartoons of SNAP-mGluR2-TMD (top) and SNAP-mGluR3-TMD (bottom) labeled with fluorophore LD555. Right, representative images showing expression and surface labeling of SNAP-mGluR2-TMD (top) and SNAP-mGluR3-TMD (bottom) in HEK 293 T cells before lysis. (**B**) Top, schematic showing the SiMPull setup. Bottom, representative image of single molecules with representative fluorescence time course for an individual protein complex (red circle) demonstrating two-step photobleaching. Scale bar = 10 µm. (**C**) Histogram summarizing the photobleaching step distribution for SNAP-mGluR2-TMD (n = 1598 total spots from 14 movies) and SNAP-mGluR3-TMD (n = 1435 total spots from 10 movies). Dashed line on the SNAP-mGluR3-TMD plot shows the normalized photobleaching step distribution for SNAP-mGluR2-TMD for comparison. (**D**) Bar graph showing percent dimerization for SNAP-tagged constructs. * indicates statistical significance (one-way ANOVA, p=8.9E-30; Tukey-Kramer for mGluR2 vs. mGluR3 p=0.78, for mGluR3 vs. mGluR2-TMD p=6.9E-8, for mGluR2-TMD vs. mGluR3-TMD p=8.2E-13, for mGluR3-TMD vs. β2AR p=1.7E-12). (**E**) Bar graph showing surface expression for constructs in (**D**). Values are normalized to SNAP-mGluR2. Expression is not significantly different between constructs (one-way ANOVA, p=0.64). (**F**) Bar graph showing the percent dimerization for SNAP-tagged constructs. * indicates statistical significance (one-way ANOVA, p=2.1E-17; Tukey-Kramer for mGluR2 vs. mGluR2-C121A p=0.27, for mGluR2-C121A vs. mGluR2-C121A-mGluR3TMD p=2.5E-5, for mGluR3 vs. mGluR3-C127A p=1.1E-12, for mGluR3-C127A vs. mGluR3-C127A-mGluR2TMD p=4.1E-8). (**G**) Bar graph showing surface expression for constructs in (**F**). Values are normalized to SNAP-mGluR2. Expression is not significantly different between constructs (one-way ANOVA, p=0.12). (**H**) Schematic illustrating the effect of inter-TMD interactions in mGluR2 and mGluR3 dimer assembly. The '<' and '~' symbols refer to relative differences in dimerization propensity. Number of movies analyzed for each condition is shown in parenthesis above each bar. Error bars are s.e.m. Associated figure supplements include *Figure 1—figure supplements 1–5*.

The online version of this article includes the following source data and figure supplement(s) for figure 1:

**Source data 1.** Data and statistics for Figure 1 and associated supplemental figures.
**Figure supplement 1.** Further characterization of single-molecule pulldown of mGluR TMDs: lack of subunit exchange in detergent.
**Figure supplement 2.** Inter-TMD dimerization propensities are maintained with C-terminal pulldown of mGluR2 and mGluR3 TMDs.
**Figure supplement 3.** Ensemble FRET dequenching measurements show higher levels of inter-TMD FRET for mGluR2 compared to mGluR3.
**Figure supplement 4.** CG MD analysis of mGluR TMD dimerization.
**Figure supplement 5.** Implied timescales as a function of lag time for CG MD simulation.

thought of as capturing a 'snapshot' of TMD dimers and monomers in the cell membrane at the time of lysis.

Consistent with prior work, SNAP-mGluR3-TMD molecules photobleached with a greater number of one-step (69.1%) events, less two-step (27.9%) events, and a similarly small population of ≥3 steps (2.9%) (*Figure 1C*) compared to SNAP-mGluR2-TMD. Thus, unlike their full-length counterparts which both show strict dimerization (*Levitz et al., 2016*), mGluR2-TMD and mGluR3-TMD show

distinct dimerization propensities that are less than an obligate dimer but greater than a canonical class A GPCR, β2AR, which we used as a monomeric or minimal dimerization control (*Figure 1D*). These dimerization differences appear to reflect truly distinct assembly properties as surface expression levels were similar across constructs (*Figure 1E*). Importantly, SNAP-mGluR2-TMD and SNAP-mGluR3-TMD retained their differential dimerization propensities when they were immobilized via their C-terminus and when SiMPull experiments were conducted under different detergent conditions (*Figure 1—figure supplement 2*).

To confirm that the dimerization propensities observed in detergent-based single-molecule subunit counting experiments reflect those occurring in the cell membrane, we measured ensemble FRET levels for the TMDs of both mGluR2 and mGluR3 at comparable expression levels in live cells. SNAP-mGluR2-TMD or SNAP-mGluR3-TMD constructs were labeled with SNAP-reactive donor (BG-LD555) and acceptor (BG-LD655) fluorophores. Fluorescence recovery of the donor was examined following acceptor bleaching with 640 nm illumination (*Figure 1—figure supplement 3A*). The percent change in donor fluorescence was greater for SNAP-mGluR2-TMD compared to SNAP-mGluR3-TMD, consistent with the interpretation that mGluR2 has a greater inter-TMD dimerization propensity compared to mGluR3 (*Figure 1—figure supplement 3B*). As a control, SNAP-β$_2$AR showed less donor recovery than both mGluR constructs (*Figure 1—figure supplement 3B*). Combined, these results indicate that our single molecule experiments are an accurate reflection of mGluR dimerization in the membrane.

To provide a structural context to the different dimerization propensities of mGluR2 and mGluR3, we used extended coarse-grained (CG) molecular dynamics (MD) simulations of pairs of receptors randomly placed relative to each other in a 1-palmitoyl-2-oleoyl-sn-glycero-3-phosphocholine (POPC) lipid bilayer (*Figure 1—figure supplement 4A*) and analyzed their most probable interface formation with Markov state models (MSMs) (*Figure 1—figure supplement 4B*). The results reveal the exploration of multiple dimeric configurations (or macrostates) for both mGluR2 and mGluR3 (*Figure 1—source data 1*) and confirm a higher dimerization propensity for mGluR2-TMD compared to mGluR3-TMD (~40% and ~25% dimeric fractions, respectively), as well as a differential involvement of TMs in dimeric interfaces (see figure legend of *Figure 2—figure supplement 2B* for list of microstates). To determine the optimal lag time and validate the Markov model, we estimated the transition probability matrix $P(\tau)$ for increasing values of the lag time $\tau$. The implied relaxation timescales predicted by each Markov model were calculated as $t_i = -\tau/\log(\lambda_i)$, where $\lambda_i$ are the eigenvalues of the transition matrix (*Figure 1—figure supplement 5*). For lag times $\tau \geq 20\mathrm{ns}$ the predicted implied timescales are approximately time-independent, indicating that the probabilities and kinetics calculated from the transition matrix at this lag time accurately reflect the behavior of the system. We grouped all sampled microstates into macrostates consisting of symmetric and asymmetric dimeric configurations with specific TM helices exhibiting high probability (larger than 40%) to be at an interface and forming the largest number of dimeric contacts (*Figure 1—figure supplement 4B-G*). This analysis draws attention to TM1, TM4, and TM5 as the most contributing helices (macrostates with probabilities larger than 10%) in the case of mGluR2 or TM3 and TM7 in the case of mGluR3 (*Figure 1—source data 1*). Significant transition fluxes observed between the different dimeric macrostates (*Figure 1—figure supplement 4B*) indicate that direct interconversion between some of the identified dimeric configurations does not require reverting first to the monomeric state.

Next, we asked whether the variable propensity for TMD dimerization between mGluR2 and 3 has an effect in the context of the full-length receptor. To sensitize our constructs to effects at non-covalent interfaces, we mutated a cysteine within the LBD that produces an inter-subunit disulfide bond. Our previous work has demonstrated that mGluRs remain functional with this mutation (*Levitz et al., 2016*). Consistent with previous work (*Levitz et al., 2016*), mutation of this intersubunit disulfide bridge in mGluR2 (C121A) had a minimal effect on mGluR2 dimerization (*Figure 1F*). However, in stark contrast, the equivalent mutation in mGluR3, C127A, produced a substantial two-fold reduction in dimerization (*Figure 1F*). We hypothesized that differences in sensitivity to cysteine mutation are due to differences in the relative inter-TMD interaction and reasoned that if the TMD plays a role in mGluR dimerization, an mGluR2-C121A chimera with the mGluR3 TMD ('mGluR2-C121A-mGluR3TMD') would reduce dimerization further. Indeed, this construct showed normal expression (*Figure 1G*), but reduced dimerization compared to mGluR2-C121A (*Figure 1F*). We next tested the reverse chimera to see if the mGluR2-TMD can rescue the reduced dimerization of

mGluR3-C127A. As hypothesized, 'mGluR3-C127A-mGluR2TMD' showed unaltered expression (*Figure 1G*) but increased dimerization compared to SNAP-mGluR3-C127A (*Figure 1F*). Together these results indicate that the TMD contributes to differences in dimerization between full-length mGluR2 and mGluR3. Specifically, the greater dimer propensity between mGluR2 TMDs increases the dimerization of full-length mGluR2 while the decreased inter-TMD interactions in mGluR3 leads to less favorable dimer formation and therefore greater dimer dissociation in the mGluR3-C127A background (*Figure 1H*). Notably, TMD chimeras were not able to fully swap the relative dimerization propensities between subtypes, suggesting that other differences outside of the TMD also contribute to differential assembly of mGluR2 and mGluR3.

## Residues in TM4 control the relative TMD dimerization propensities of mGluR2 and mGluR3

Having demonstrated that the TMD plays a critical role in subtype-specific modes of mGluR dimerization, we wondered which specific regions of the TMD contribute to this effect. Together the existing literature suggests that inter-TMD interaction among class C GPCRs is multifaceted and diverse (see Introduction), motivating further analysis and careful comparison between receptor subtypes. We started by assessing sequence conservation across mGluR subtypes. Across all eight rat mGluRs, most TM helices show <50% sequence conservation and <30% sequence identity, while TM6 stands out as the most similar, with 96% conservation and 72% identity (*Figure 2—figure supplement 1A*). Analysis of only the group II mGluRs reveals much greater sequence similarity, with all TMs having >80% conservation. Restricting analysis to outward-facing residues (defined by homology models based on the mGluR5 TMD crystal structure), which we expect to disproportionately contribute to TMD interface formation, we see marked differences (*Figure 2—figure supplement 1A–C*). Across all mGluRs, only TM6 remains highly similar, with 87.5% conservation and 87.5% identity. Between the group II mGluRs, conservation remains high (>70%) for all TMs except for TM4 (33%) (*Figure 2B*). Outward-facing residues in TM4 also exhibited the lowest (0%) sequence identity of the group II mGluR TMs. Given that the overall sequence identity between mGluR2 and mGluR3 is nearly 70%, it is reasonable to hypothesize that the increased heterogeneity of TM4 contributes to subtype-specific differences in TMD dimerization.

Our CG MD simulations showed a range of mGluR2 and mGluR3 TMD dimerization interfaces (*Figure 1—figure supplement 4*). However, we hypothesized that the presence of large, dimeric extracellular domains might give preference to specific dimerization modes. To test this hypothesis, we carried out steered MD simulations of full-length CG models to representative dimeric configurations of the two receptors and calculated energetic differences between them (*Figure 2—figure supplement 2*). Notably, while the identified symmetric TM1/TM1 dimeric configuration of the mGluR2-TMD was suggested to be more stable than the asymmetric TM1/TM4 and TM1/TM5 mGluR2-TMD dimers (energy differences of $-580 \pm 18$ kJ mol$^{-1}$, $-169 \pm 18$ kJ mol$^{-1}$, respectively) in the presence of the LBD, other dimeric configurations such as the symmetric (TM4)/(TM4) dimer of mGluR2 (*Figure 2—figure supplement 3A*), showed higher compatibility with the full-length receptor than the TM1/TM1 dimer and other highly populated macrostates (energy differences of $-1370 \pm 18$ kJ mol$^{-1}$, $-1950 \pm 18$ kJ mol$^{-1}$, and $-1540 \pm 18$ kJ mol$^{-1}$ from the TM1/TM1, TM1/TM4, and TM1/TM5, respectively). In contrast, the identified TM4/TM4 dimer configuration of mGluR3-TMD (*Figure 2—figure supplement 3B*), was estimated to be less stable than the identified largest mGluR3-TMD dimer macrostate in the presence of the LBD (energy difference of $+630 \pm 12$ kJ mol$^{-1}$ from the TM7/TM3 interface).

Together, examination of amino acid sequence conservation and CG MD simulations point to TM4 as a potential mediator of the differential TMD dimerization of mGluR2 and mGluR3. Intriguingly, differences in TM4 between the group II mGluRs have been reported to contribute to the specific interaction of mGluR2, but not mGluR3, with the 5-HT$_{2A}$ serotonin receptor, a class A GPCR (*González-Maeso et al., 2008*; *Fribourg et al., 2011*; *Moreno et al., 2016*; *Delille et al., 2012*). This interaction has been attributed to three outward-facing alanine residues (A677, A681, A685) at the cytosolic end of TM4 of mGluR2 (*Figure 2A*) that allow mGluR2, but not mGluR3, to interact with 5-HT$_{2A}$R (*Moreno et al., 2012*). Importantly, these three alanine residues and the corresponding residues in mGluR3 are conserved across species (*Figure 2—figure supplement 1D*) and are observed within many of the TM4-containing dimer interfaces obtained by CG MD simulations

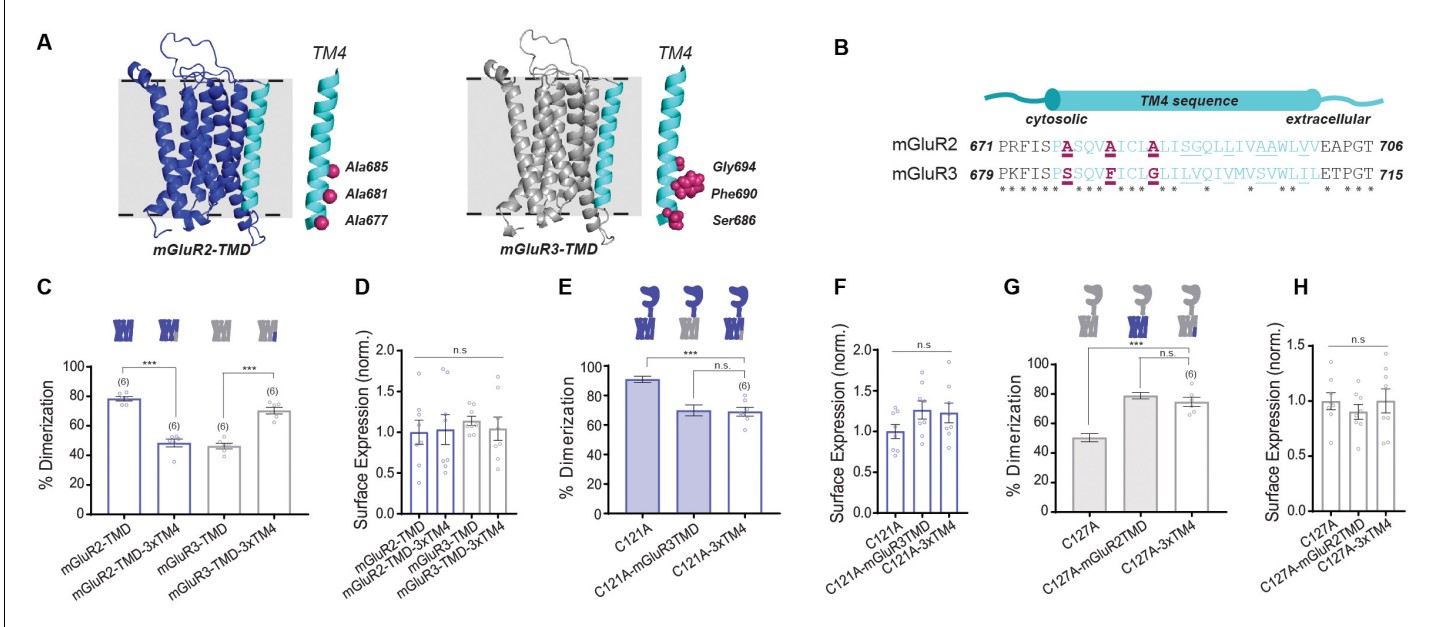

**Figure 2.** Residues in TM helix four mediate differences in inter-TMD interaction between mGluR2 and mGluR3. (A) Homology models of the TMD of mGluR2 (purple) and mGluR3 (gray) showing TM4 (cyan) residues of interest in magenta. (B) TM4 sequence alignment for mGluR2 and mGluR3. (C) Bar graph showing percent dimerization for SNAP-tagged TMD constructs. * indicates statistical significance (one-way ANOVA, p=4.7E-10; Tukey-Kramer for mGluR2-TMD vs. mGluR2-TMD-3xTM4 p=1.5E-8, for mGluR3-TMD vs. mGluR3-TMD-3xTM4 p=5.3E-7). (D) Bar graph showing surface expression for constructs in (C). Values are normalized to SNAP-mGluR2-TMD. Expression is not significantly different between constructs (one-way ANOVA, p=0.91). (E) Bar graph showing percent dimerization for SNAP-tagged full-length constructs. * indicates statistical significance (one-way ANOVA, p=7.6E-5; Tukey-Kramer for mGluR2-C121A vs. mGluR2-C121A-3xTM4 p=0.00015, for mGluR2-C121A-mGluR3TMD vs. mGluR2-C121A-3xTM4 p=0.98). Shaded bars show repeated data from *Figure 1F*. (F) Bar graph showing surface expression for constructs in (E). Values are normalized to SNAP-mGluR2-C121A. Expression is not significantly different between constructs (one-way ANOVA, p=0.19). (G) Bar graph showing percent dimerization SNAP-tagged constructs labeled with LD555. * indicates statistical significance (one-way ANOVA, p=3.6E-7; Tukey-Kramer for mGluR3-C127A vs. mGluR3-C127A-3xTM4 p=1.5E-5, for mGluR3-C127A-mGluR2TMD vs. mGluR3-C127A-3xTM4 p=0.53). Shaded bars show repeated data from *Figure 1F*. (H) Bar graph showing surface expression for constructs in (G). Values are normalized to SNAP-mGluR3-C127A. Expression is not significantly different between constructs (one-way ANOVA, p=0.65). Number of movies analyzed for each condition is shown in parenthesis above each bar. Error bars are s.e.m. Associated figure supplements include *Figure 2—figure supplements 1–6*.

The online version of this article includes the following source data and figure supplement(s) for figure 2:

**Source data 1.** Data and statistics for Figure 2 and associated supplemental figures.

**Figure supplement 1.** mGluR TMD sequence conservation analysis.

**Figure supplement 2.** Convergence assessment of the free energy differences between representative dimeric configurations of highly populated macrostates.

**Figure supplement 3.** Symmetric TM4-TM4 interfaces for group II mGluR TMDs.

**Figure supplement 4.** Further analysis of dimerization of TM4 mutants of mGluR2 and mGluR3 TMD constructs.

**Figure supplement 5.** Further analysis of mutations to outward-facing mGluR2 TMD residues: TM1 and TM5 mutants do not alter dimerization .

**Figure supplement 6.** Further analysis of dimerization of mGluR2 and mGluR3 mutant constructs.

(*Figure 2—figure supplement 3*). We thus wondered whether these alanine residues also play a role in mGluR homodimer formation.

To test our hypothesis, we swapped A677, A681, and A685 in mGluR2 with the corresponding residues in mGluR3: S686, F690, and G694 (*Figure 2A,B*) and vice versa to create two mGluR-TMD mutants, 'mGluR2-TMD-3xTM4' and 'mGluR3-TMD-3xTM4'. Strikingly, we found that dimerization propensity was determined by which 3xTM4 residues were present in the construct (*Figure 2C*). For mGluR2-TMD-3xTM4, we observed a reduction in dimerization from 78.4 ± 1.4% for mGluR2-TMD to 48.4 ± 2.4%. Conversely, mGluR3-TMD-3xTM4 increased dimerization from 46.3 ± 1.7% for mGluR3-TMD to 70.0 ± 2.0%. These data indicate that TM4 residues determine the relative dimerization propensity of group II mGluRs at the TMD level. Importantly, both mutants expressed similarly to wild-type receptors (*Figure 2D*). Single amino acid swap constructs in the SNAP-mGluR2-TMD

background show that although each mutation individually alters dimerization, all three mutants combined in the 3xTM4 constructs are required for the greatest effect (*Figure 2—figure supplement 4*).

We also examined whether mutating other non-conserved outward-facing residues in TM4 influences TMD dimerization. We swapped mGluR2 residues A695 and A696 with mGluR3 residues S703 and V704 to make mGluR2-TMD-A695S/A696V. Notably, these positions were shown to cross-link in a prior study of mGluR2 (*Xue et al., 2015*). mGluR2-TMD-A695S/A696V exhibited indistinguishable expression and dimerization compared to SNAP-mGluR2-TMD, suggesting that outward-facing residues at the extracellular end of TM4 do not influence the relative TMD dimerization propensities between mGluR2 and mGluR3 (*Figure 2—figure supplement 5A–C*). We also tested differences between mGluR2 and mGluR3 in TM1 (F584, G587 in mGluR2 and I593, T596 in mGluR3) and TM5 (A726, G730, A733 in mGluR2 and S735, I739, T742 in mGluR3) by swapping non-conserved outward-facing residues and observed no effect in SiMPull or in terms of surface expression (*Figure 2—figure supplement 5D–G*). Among these residues, only one (F584 in mGluR2 and I593 in mGluR3) was found to participate in dimeric configurations identified by simulation (specifically, the TM1/TM1 interface; *Figure 1—figure supplement 4*). While these SiMPull and computational results do not rule out the possibility that the extracellular end of TM4, TM1 or TM5 contribute to TMD dimerization, they show that these regions are not major mediators of the differences between mGluR2 and mGluR3.

While isolated TMDs are very useful for studying TMD behavior directly, we asked if the identified TM4 residues are relevant within the context of full-length mGluRs. We first produced an mGluR2-C121A-3xTM4 construct and examined its dimerization propensity compared to mGluR2-C121A and mGluR2-C121A-mGluR3TMD. Consistent with isolated TMD data, we found that the effect of swapping the entire TMD was mimicked by only introducing the 3xTM4 mutant (*Figure 2E*). This effect was also observed with mGluR3-C127A-3xTM4 where the increase in dimerization matched what was observed for mGluR3-C127A-mGluR2TMD (*Figure 2G*). All mutants showed no effect on receptor expression levels (*Figure 2F,H*).

Together, these experiments show that TM4 residues control the relative TMD dimerization propensity, at least for mGluR2. Importantly, TMD chimeras or 3xTM4 swap mutations showed no clear effect on dimerization in full-length wild type backgrounds likely due to the maintenance of the disulfide bridge that stabilized LBD dimerization (*Figure 2—figure supplement 6*). This indicates that the cysteine mutation is needed to reveal the distinct inter-TMD interaction modes that exist within mGluR2 and mGluR3 dimers.

## TM4 residues control orthosteric and allosteric activation of group II mGluRs

Having observed a clear, but variable role for inter-TMD interfaces involving the cytosolic end of TM4 in dimerization, we next investigated how this region contributes to mGluR function. We turned to a calcium imaging assay in HEK 293 T cells where a G protein chimera (*Conklin et al., 1993*) allows $G_{i/o}$-coupled receptors, such as mGluR2 and mGluR3, to produce release of calcium from intracellular stores which can be quantified by measuring the fluorescence increase from a fluorescent calcium sensor such as GCaMP6f (Materials and methods). We started by measuring the glutamate response of wild type mGluR2 and mGluR3. As expected from previous studies (*Levitz et al., 2016*; *Tora et al., 2018*), mGluR3 showed a higher apparent glutamate affinity compared to mGluR2 (*Figure 3—figure supplement 1A,B*). Disrupting the TM4 interface in mGluR2-3xTM4 produced a four-fold leftward shift compared to wild-type mGluR2 (*Figure 3A–C*). Based on this result, we wondered whether mGluR3-3xTM4 would have reduced apparent glutamate affinity compared to mGluR3. Indeed, we observed a 20-fold rightward shift for mGluR3-3xTM4 compared to wild type mGluR3 (*Figure 3D–F*). To further understand the contribution of each residue, we also tested single mutants. Interestingly, mGluR2-A677S, -A681F, and -A685G did not show different glutamate $EC_{50}$ values compared to wild-type mGluR2 (*Figure 3C*; *Figure 3—figure supplement 1C*). Conversely, mGluR3-S686A, -F690A, and -G684A all exhibited significant rightward shifts compared to mGluR3, although no single mutant had as reduced apparent affinity as mGluR3-3xTM4 (*Figure 3F*; *Figure 3—figure supplement 1D*). Similar rightward and leftward shifts were observed for mGluR2-3xTM4 and mGluR3-3xTM4 using a G-protein-coupled inward-rectifier potassium (GIRK) channel-based patch-clamp assay (Materials and methods; *Figure 3—figure supplement 1E–G*), showing

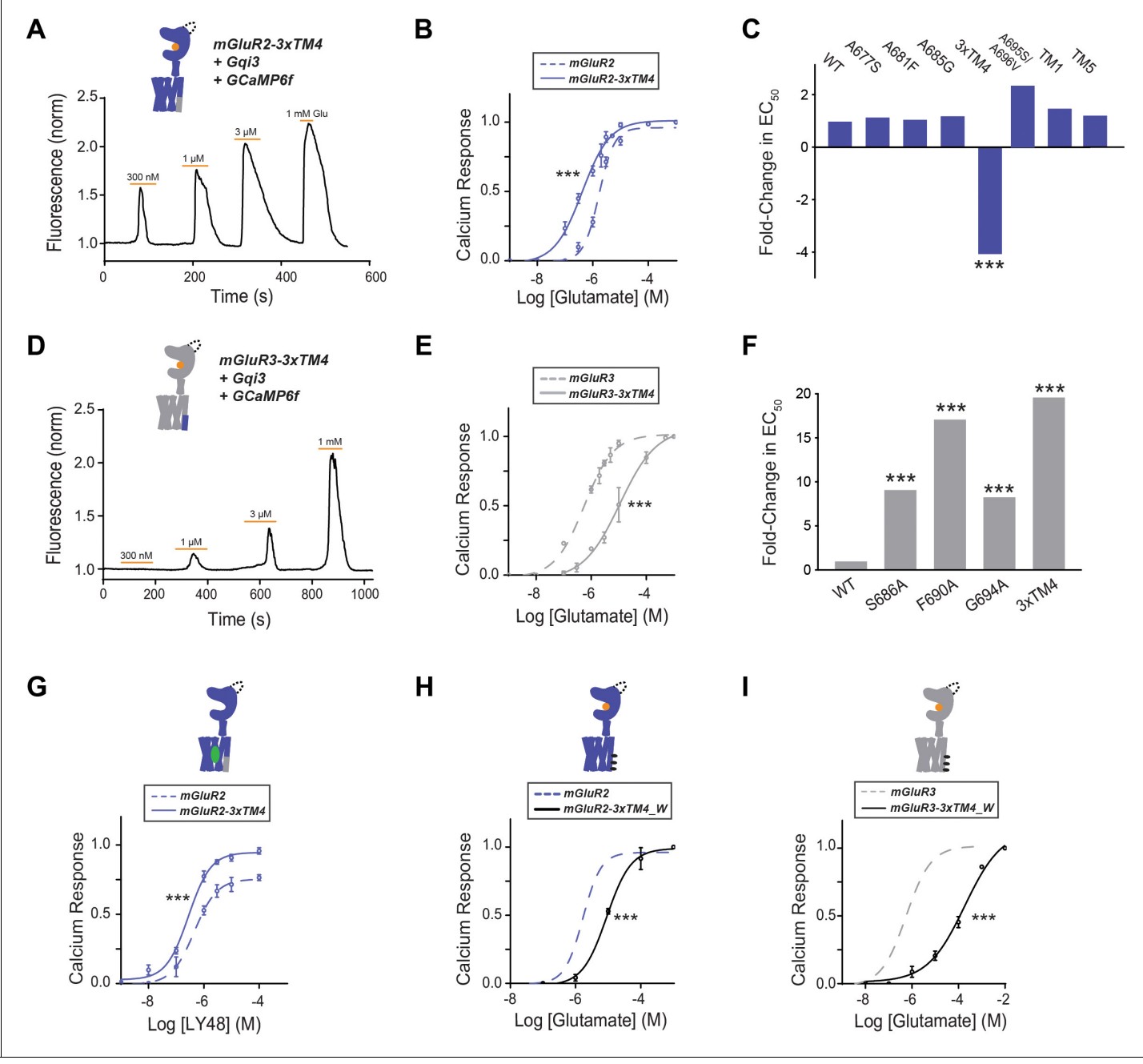

**Figure 3.** Differences in TM4 residues mediate different activation properties of mGluR2 and mGluR3. (**A**) Representative calcium imaging trace showing calcium responses induced by glutamate application from a HEK 293 T cell expressing mGluR2-3xTM4, a G-protein chimera and GCaMP6f. (**B**) Glutamate calcium imaging concentration response curves for mGluR2 ($EC_{50} = 1.68 \pm 0.03$ µM) and mGluR2-3xTM4 ($EC_{50} = 0.41 \pm 0.06$). All values are normalized to saturating (1 mM) glutamate. * indicates statistical significance (two-way ANOVA, p=5.2E-10). (**C**) Summary of glutamate $EC_{50}$ shifts relative to wild-type mGluR2 obtained for mGluR2 mutants. * indicates statistical significance (two-way ANOVA; for mGluR2 vs. mGluR2-3xTM4, p=5.2E-10). 'TM1'=F584I, G587T; 'TM5'=A726S, G730I, A733T (**D**) Representative calcium imaging trace showing calcium responses from mGluR3-3xTM4. (**E**) Glutamate calcium imaging concentration response curves for mGluR3 ($EC_{50} = 0.56 \pm 0.06$) and mGluR3-3xTM4 ($EC_{50} = 11.0 \pm 0.1$). All values are normalized to saturating (1 mM) glutamate. * indicates statistical significance (two-way ANOVA, p=3.6E-10). (**F**) Summary of glutamate $EC_{50}$ shifts relative to wild-type mGluR3 obtained for mGluR3 mutants. * indicates statistical significance (two-way ANOVA; for mGluR3 vs. mGluR3-S686A, p=2.2E-8; for mGluR3 vs. mGluR3-F690A, p=2.6E-14; for mGluR3 vs. mGluR3-G694A, p=4.8E-10; for mGluR3 vs. mGluR3-3xTM4, p=3.6E-10). (**G**) LY48 calcium imaging concentration response curves for mGluR2 ($EC_{50} = 0.45 \pm 0.10$) and mGluR2-3xTM4 ($EC_{50} = 0.27 \pm 0.06$). mGluR2-3xTM4 has ~17% greater efficacy than mGluR2 at saturating LY48. All values are normalized to saturating (1 mM) glutamate. * indicates statistical significance (two-way ANOVA, p=1.3E-7). (**H**) Glutamate calcium imaging concentration response curves for mGluR2 and mGluR2-3xTM4_W ($EC_{50} = 9.01 \pm 0.08$). All values are

*Figure 3 continued on next page*

*Figure 3 continued*

normalized to saturating (1 mM) glutamate. * indicates statistical significance (two-way ANOVA, p=0.00087). (I) Glutamate calcium imaging concentration response curves for mGluR3 and mGluR3-3xTM4_W (EC$_{50}$ = 186.00 ± 0.27). All values are normalized to saturating (1 mM) glutamate. * indicates statistical significance (two-way ANOVA, p=1.2E-10). Associated figure supplement includes *Figure 3—figure supplements 1–2*.

The online version of this article includes the following source data and figure supplement(s) for figure 3:

**Source data 1.** Data and statistics for Figure 3 and associated supplemental figures.
**Figure supplement 1.** Further data showing differences in TM4 residues mediate different activation properties of mGluR2 and mGluR3.
**Figure supplement 2.** Further analysis of 3xTM4_W mutants in mGluR2 and mGluR3 constructs.

that the effects of TM4 mutations on function are maintained using endogenous G$_{i/o}$ proteins. Together, these measurements show that inter-TM4 interactions influence orthosteric activation of group II mGluRs.

We also tested glutamate responses of the TM1 (F584I/G587T), TM4 (A695S/A696V) and TM5 (A726S/G730I/A733T) mutants which showed no change compared to wild-type in dimerization experiments (*Figure 2—figure supplement 5*). Consistent with their lack of effect on dimerization, all three mutants showed identical concentration response curves compared to wild-type mGluR2 (*Figure 3C*; *Figure 3—figure supplement 1I*).

Next, we wondered whether the 3xTM4 residues also influence allosteric activation. Our work and others' have demonstrated that positive allosteric modulators (PAMs) can function as direct agonists of mGluR2 (*Gutzeit et al., 2019*; *El Moustaine et al., 2012*; *Goudet et al., 2004*; *O'Brien et al., 2018*). We tested the mGluR2 PAM LY487379 (LY48) (*Johnson et al., 2003*) with mGluR2 and mGluR2-3xTM4. Like the glutamate response, we observed a small leftward shift in EC$_{50}$ for mGluR2-3xTM4 compared to wild type mGluR2 (*Figure 3G*, *Figure 3—figure supplement 1H*). We also observed that the mGluR2-3xTM4 LY48 PAM response showed greater maximal efficacy, with saturating (100 µM) LY48 giving 19.4% greater amplitude compared to wild type mGluR2. This finding shows that 3xTM4 residues also influence allosteric activation.

Given the clear effects of TM4 perturbations on group II mGluR function, we asked what the effects of harsher alteration of this interface would be. We tested this by introducing bulky, hydrophobic tryptophan residues into outward facing residues of TM4 to disrupt inter-subunit interaction. We first produced mGluR2-3xTM4_W (A677W, A681W, A685W) in the isolated TMD background and observed reduced dimerization compared to both mGluR2-TMD and mGluR2-3xTM4 (*Figure 3—figure supplement 2A*), despite normal surface expression levels (*Figure 3—figure supplement 2B*). In contrast to what was observed with mGluR2-3xTM4, full-length mGluR2-3xTM4_W showed a small right-shift in the functional calcium imaging concentration response curve (*Figure 3H*), suggesting that harsh perturbation to the lipid-targeting face of TM4 impairs activation. Consistent with this, mGluR3-3xTMD_W (S686W/F690W/G684W) showed similarly reduced dimerization compared to wild-type mGluR3-TMD (*Figure 3—figure supplement 2A*) despite normal surface expression levels (*Figure 3—figure supplement 2B*) and showed a pronounced right-shift compared to wild-type mGluR3 in the functional calcium imaging assay (*Figure 3I*). Together these results confirm the importance of TM4 in group II mGluR TMD dimerization and suggest that outward-facing residues in TM4 influence inter-subunit rearrangement for both receptor subtypes.

## SNAP- and Halo-based FRET sensors enable analysis of the effects of TM4 interface perturbation on conformational dynamics of full-length mGluRs

Having established the role of TM4 residues in influencing dimerization and both orthosteric and allosteric activation, we next wondered how they influence mGluR conformational dynamics using fluorescence resonance energy transfer (FRET). We first examined whether TM4 residues influence inter-LBD dynamics using a well-established assay where a decrease in FRET between N-terminal SNAP-tags is correlated with activation-associated inter-LBD reorientation (*Doumazane et al., 2013*; *Olofsson et al., 2014*; *Vafabakhsh et al., 2015*). HEK 293 T cells were transfected with SNAP-tagged mGluR constructs and after labeling with SNAP-reactive donor (BG-LD555) and acceptor (BG-LD655) fluorophores (Materials and methods), the donor fluorophore was excited with a 561 nm laser while donor and acceptor channels were imaged simultaneously with a dual camera imaging

system as described previously (*Gutzeit et al., 2019*). 3xTM4 mutants produced inter-LBD FRET responses that were indistinguishable from their wild-type counterparts with identical glutamate concentration-response curves (*Figure 4—figure supplement 1A,B*). As previously shown, mGluR3 but not mGluR2 showed a clear basal FRET response to the competitive antagonist LY341495 in the absence of glutamate, consistent with constitutive inter-LBD dynamics that drive basal activity (*Vafabakhsh et al., 2015*; *Levitz et al., 2016*). mGluR3-3xTM4 maintained a basal response to LY34 that was indistinguishable from wild-type, while both mGluR2 and mGluR2-3xTM4 showed no basal response to LY34 (*Figure 4—figure supplement 1C,D*). Together these data show that 3xTM4 mutants undergo normal LBD motions, suggesting that their functional effects are mediated downstream at the TMDs themselves. Furthermore, this underscores the large effects of the 3xTM4 mutations on receptor activation, given that despite normal glutamate sensitivity at the LBDs apparent glutamate affinities were shifted by up to nearly 20-fold in the case of mGluR3 (*Figure 3*).

We next sought to directly probe dynamics at the TMD by developing new FRET sensors based on ours and others' previous studies of mGluR1 where insertion of fluorescent proteins (i.e. CFP/YFP) into intracellular loops or C-termini have yielded sensors of activation-associated conformational changes that are suitable for live cell studies (*Hlavackova et al., 2012*; *Marcaggi et al., 2009*; *Grushevskyi et al., 2019*; *Tateyama et al., 2004*; *Tateyama and Kubo, 2006*). We re-engineered this approach to incorporate HaloTags (*Los et al., 2008*; *Schihada et al., 2018*) which enable the use of organic fluorophores, such as the family of rhodamine-based Janelia Fluor ('JF') dyes (*Grimm et al., 2016*; *Grimm et al., 2020*; *Grimm et al., 2015*), which show enhanced brightness and stability compared to fluorescent proteins. We produced constructs where a HaloTag is inserted into the second intracellular loop of mGluR2 or mGluR3 to monitor proximity between TMDs (*Figure 4A*). It is important to note that these constructs are unable to couple to G proteins as has been reported with all previous inter-TMD fluorescent-protein based sensors of full-length mGluRs (*Tateyama et al., 2004*; *Tateyama and Kubo, 2006*; *Marcaggi et al., 2009*; *Hlavackova et al., 2012*; *Grushevskyi et al., 2019*). Following expression, these constructs were labeled with HaloTag donor ($JF_{549}$) and acceptor ($JF_{646}$) dyes at a 1:3 ratio where optimal results were observed (*Figure 4—figure supplement 2A*). Consistent with prior fluorescent protein-based constructs for mGluR1, glutamate application resulted in a FRET increase for both the mGluR2 and mGluR3 sensors (*Figure 4B*; *Figure 4—figure supplement 2B,C*). Importantly, no FRET responses were observed when constructs were labeled with only donor or only acceptor fluorophores (*Figure 4—figure supplement 2D,E*). FRET responses were reversible upon drug washout and concentration-dependent (*Figure 4—figure supplement 3A,B*). Concentration response curves for wild-type constructs showed the expected higher apparent glutamate affinity for mGluR3 compared to mGluR2 (*Figure 4C*).

We further characterized the sensitivity of the FRET sensors to different types of compounds. As expected, glutamate responses were fully blocked by the competitive antagonist LY341495 (*Figure 4—figure supplement 3C,D*) at the same concentration (5 µM) used for block of functional responses (*Figure 4—figure supplement 3E,F*). We next asked if glutamate-induced FRET increases are blocked by TMD-targeting mGluR NAMs. We focused on the non-specific group II mGluR NAM MNI 137, which (10 µM) was only able to produce a weak, partial block of glutamate-induced FRET responses (32.0 ± 4.5% for mGluR2; 42.9 ± 7.4% for mGluR3) (*Figure 4D,E*; *Figure 4—figure supplement 3G*) at the same saturating concentration (10 µM) where full block of functional responses was observed (*Figure 4E*; *Figure 4—figure supplement 3H,I*). Further raising the MNI 137 concentration to 50 µM ($IC_{50}$ ~10 nM *Hemstapat et al., 2007*) did not increase the extent of block of glutamate-induced mGluR3 FRET responses (44.9 ± 8.3%; n = 3). These results indicate that these FRET sensors partially report on conformational changes that are insensitive to NAMs. In addition, we tested a saturating concentration of the mGluR2 PAM LY48 and, in contrast to its robust functional activation properties (*Figure 3G*), observed weak FRET increases that were 33.5 ± 3.9% of the amplitude of the response to saturating glutamate (*Figure 4F*).

Next, we asked if the inter-TMD FRET sensors are sensitive to perturbation to TM4. We found that mGluR2-3xTM4 exhibited a clear leftward-shift compared to mGluR2, while mGluR3-3xTM4 exhibited a rightward-shift compared to mGluR3 (*Figure 4G–I*), supporting a role for TM4 in controlling intersubunit motions. In addition, mGluR3-3xTM4 exhibited a reduced basal FRET response to LY341495 compared to wild type mGluR3 (*Figure 4J,K*), suggesting that the increased dimerization propensity of the TM4 interface is capable of decreasing the basal activity of the receptor. We

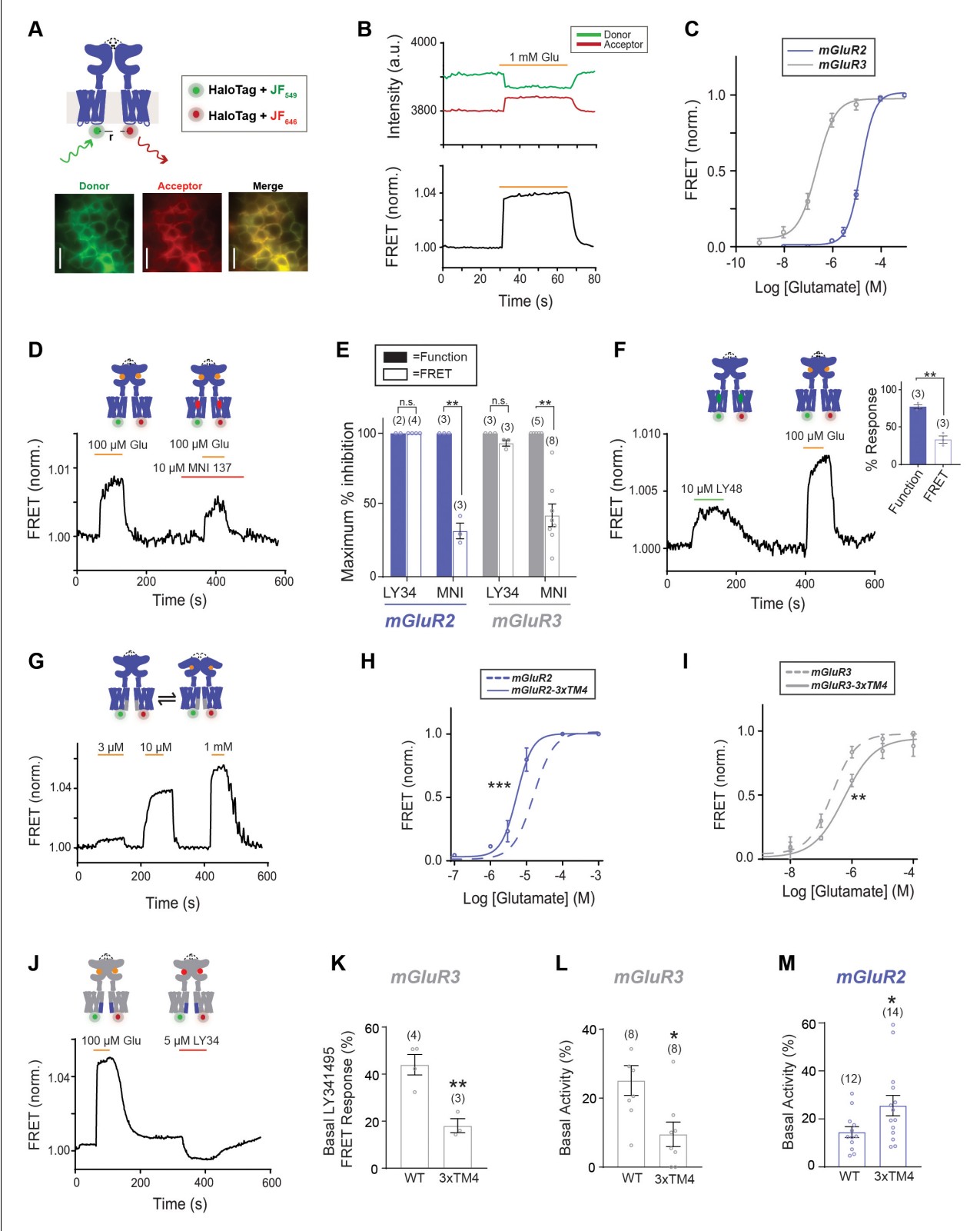

**Figure 4.** An inter-TMD FRET assay enables detection of inter-TMD conformational dynamics in full-length mGluR2 and mGluR3 in live cells. (**A**) Top, schematic showing an mGluR2 dimer with a HaloTag inserted into intracellular loop 2 (ICL2). and labeled with donor and acceptor fluorophores. Bottom, representative cell images showing donor and acceptor channels after donor excitation with a 561 nm laser. Scale bar = 20 μm. (**B**) Representative FRET trace showing donor and acceptor fluorescence intensity (top) during glutamate application. A corresponding normalized FRET

*Figure 4 continued on next page*

*Figure 4 continued*

trace (bottom) shows reversible FRET increases upon glutamate stimulation. (**C**) Glutamate FRET concentration response curves for mGluR2 ($EC_{50}$ = 15.75 ± 0.04) and mGluR3 ($EC_{50}$ = 0.23 ± 0.08). All values are normalized to saturating (1 mM) glutamate. (**D**) Application of the negative allosteric modulator MNI 137 partially blocks glutamate-induced FRET increases for mGluR2. (**E**) Quantification of saturating LY34 (5 µM) and MNI (10 µM) blockade of glutamate responses in FRET and calcium imaging for mGluR2 and mGluR3. * indicates statistical significance (unpaired t-test, for MNI inhibition in FRET vs. function for mGluR2, p=0.0064; for MNI inhibition in FRET vs. function for mGluR3, p=0.013). (**F**) Representative FRET trace shows application of positive allosteric modulator LY48 at a saturating concentration induces a smaller FRET response than saturating glutamate for mGluR2. Inset, quantification of LY48 response in FRET and calcium imaging for mGluR2. * indicates statistical significance (unpaired t-test, p=0.0033) (**G**) Representative FRET trace showing glutamate titration for mGluR2-3xTM4. (**H**) Glutamate FRET concentration response curves for mGluR2 and mGluR2-3xTM4 ($EC_{50}$ = 5.39 ± 0.03). All values are normalized to saturating (1 mM) glutamate. * indicates statistical significance (two-way ANOVA, p value = 1.7E-6). (**I**) Glutamate FRET concentration response curves for mGluR3 and mGluR3-3xTM4 ($EC_{50}$ = 0.56 ± 0.10). All values are normalized to saturating (1 mM) glutamate. * indicates statistical significance (two-way ANOVA, p value = 0.0083). (**J**) Representative FRET trace showing weak sensitivity to LY34 in the absence of glutamate for mGluR3-3xTM4. (**K**) Summary showing decrease in basal FRET response to LY34 for mGluR3-3xTM4 compared to wild-type mGluR3. * indicates statistical significance (unpaired t-test, p=0.0048). (**L**) Summary showing decrease in basal activity as assessed by LY34 application in patch clamp recordings for mGluR3-3xTM4 compared to wild-type mGluR3. * indicates statistical significance (unpaired t-test, p=0.015). (**M**) Summary showing increase in basal activity as assessed by Ro 64 application in patch clamp recordings for mGluR2-3xTM4 compared to wild-type mGluR2. * indicates statistical significance (unpaired t-test, p=0.033). The number of cells tested are shown in parentheses. Error bars represent s.e.m. Associated figure supplements include ***Figure 4—figure supplements 1–4***.

The online version of this article includes the following source data and figure supplement(s) for figure 4:

**Source data 1.** Data and statistics for Figure 4 and associated supplemental figures.

**Figure supplement 1.** Differences in TM4 residues do not alter inter-LBD conformational dynamics.

**Figure supplement 2.** Characterization of inter-TMD FRET sensor in mGluR2 and mGluR3.

**Figure supplement 3.** Characterization of inter-TMD conformational dynamics in response to orthosteric and allosteric drugs.

**Figure supplement 4.** Further characterization of inter-TMD conformational dynamics and control of basal activity.

---

tested basal receptor activity using the GIRK channel patch clamp assay and, indeed, observed reduced basal activity for mGluR3-3xTM4 compared to wild-type mGluR3 (***Figure 4L***; ***Figure 4—figure supplement 4B,C***). Interestingly the basal FRET response was larger for LY341495 than MNI 137, despite comparable responses to both antagonists in the patch clamp assay (***Figure 4—figure supplement 4A,D***), further supporting the partial sensitivity of the FRET sensor to allosteric compounds. Based on the clear effect of TM4 mutations on mGluR3 basal activity, we also asked if TM4 perturbations alter basal activity of mGluR2. Since mGluR2 does not show basal LBD dynamics (***Figure 4—figure supplement 1C,D***), we used the NAM Ro 64–2259 to detect mGluR2 basal activity due to its ability to serve as an inverse agonist (***Gutzeit et al., 2019***). Using the patch clamp assay, we observed a significant increase in basal activity for mGluR2-3xTM4 compared to mGluR2 (***Figure 4M***; ***Figure 4—figure supplement 4E,F***). Together these results indicate that orthosteric and allosteric compounds influence inter-TMD conformational dynamics to control receptor activity in a way that is tuned by a TM4-containing dimer interface.

## Testing an inter-TMD rearrangement model of mGluR activation

Previous biochemical and structural studies have strongly argued that TMDs undergo reorientation to a TM6 interface upon activation (***Xue et al., 2015***; ***Koehl et al., 2019***). Notably, TM6 is highly conserved across mGluR subtypes compared to other transmembrane helices and is 100% identical between mGluR2 and mGluR3 (***Figure 2—figure supplement 1***). This supports a model where divergent inter-TMD interaction modes across mGluR subtypes in inactive states converge on a common active-state interface. To test this model of mGluR activation, we introduced mutations at the TM6 interface based on the full-length mGluR5 agonist- and PAM-bound cryo-EM structure where a single point of contact exists between TMDs centered on conserved isoleucine 791 (I779 in mGluR2 and I788 in mGluR3) in TM6 (***Koehl et al., 2019***; ***Figure 5A***). We first tested alanine (I779A) and tryptophan (I779W) mutants in the isolated mGluR2-TMD to determine if either mutation altered dimerization in SiMPull experiments. Interestingly, both mutants showed reduced dimerization compared to the wild type mGluR2-TMD (***Figure 5—figure supplement 1A***), suggesting that both TM4- and TM6-containing interfaces contribute to the dimer population in this assay. Mutation to a nearby conserved proline (P778) in the TM6 interface also decreased TMD dimerization (***Figure 5—figure supplement 1A***) and mutation to the same conserved TM6 residue in mGluR3, I788A, produced a

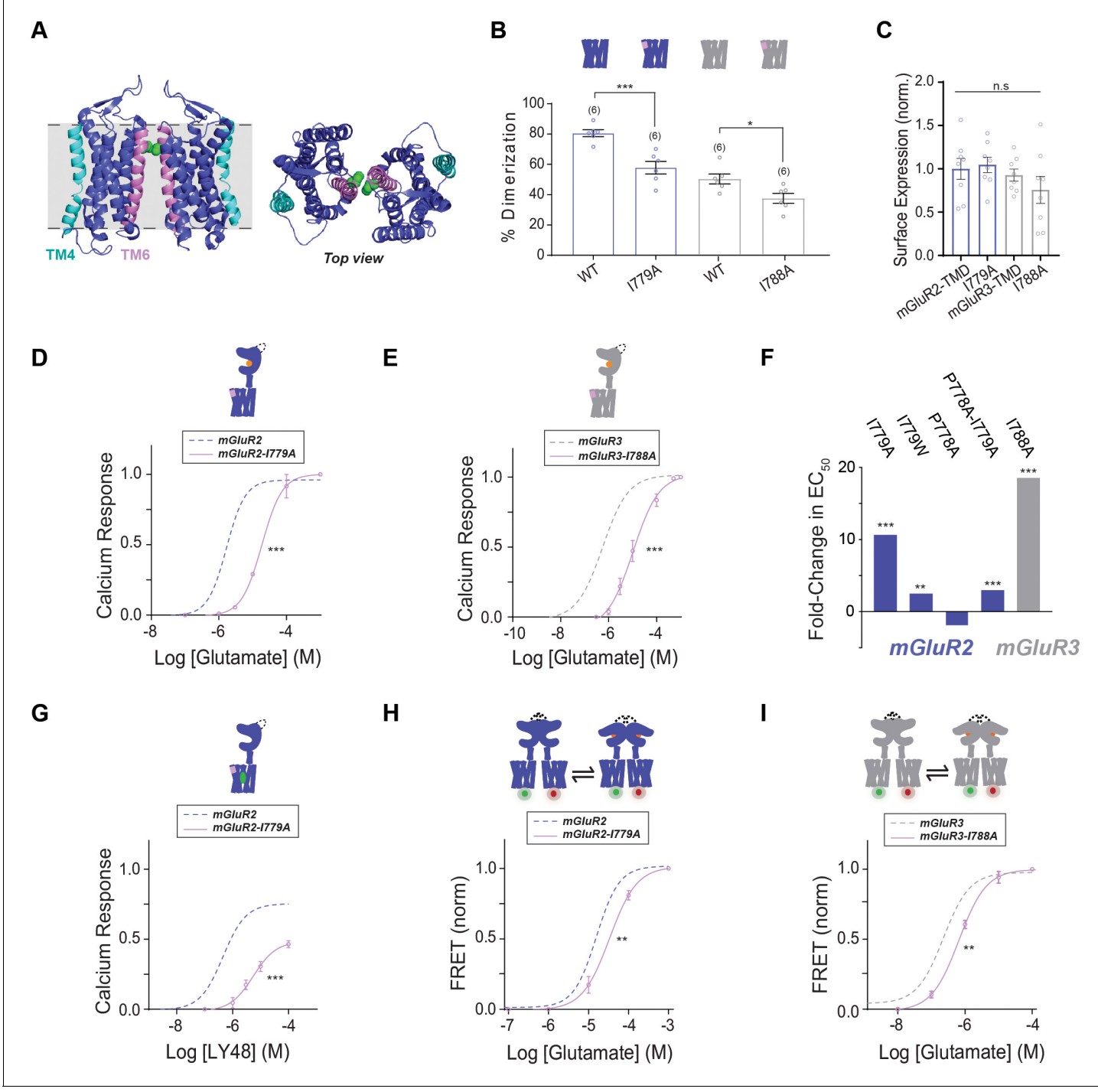

**Figure 5.** Testing the role of TM6 in group II mGluR activation. (A) Left, cryo-EM structure of mGluR5 TMD dimer (PDB: 6N51) with TM4 highlighted in cyan and TM6 highlighted in pink. The side chain of I791 (I779 in mGluR2, I788 in mGluR3) is shown in green. Right, top view. (B) Bar graph showing percent dimerization for SNAP-tagged constructs labeled with LD555. * indicates statistical significance (one-way ANOVA, p=1.2E-8; Tukey-Kramer for mGluR2-TMD vs. mGluR2-TMD-I779A p=0.00017, for mGluR3-TMD vs. mGluR3-TMD-I788A p=0.045). (C) Bar graph showing surface expression for constructs in (B). Values are normalized to SNAP-mGluR2-TMD. Expression is not significantly different between constructs (one-way ANOVA, p=0.30). (D) Calcium imaging glutamate concentration response curves for mGluR2 and mGluR2-I779A (EC$_{50}$ = 18.66 ± 0.80). All values are normalized to saturating (1 mM) glutamate. * indicates statistical significance (two-way ANOVA, p=1.2E-10). (E) Glutamate calcium imaging concentration response curves for mGluR3 and mGluR3-I788A (EC$_{50}$ = 10.43 ± 0.10). All values are normalized to saturating (1 mM) glutamate. * indicates statistical significance (two-way ANOVA, p=8.2E-10). (F) Summary of glutamate EC$_{50}$ shifts relative to wild type mGluR2 or mGluR3 obtained for their respective mutants. * indicates statistical significance (two-way ANOVA; for mGluR2 vs. mGluR2-I779A, p=1.2E-10; for mGluR2 vs. mGluR2-I779W, p=0.0028; for mGluR2 vs.

*Figure 5 continued on next page*

Figure 5 continued

mGluR2-P778A-I779A, p=0.0015; for mGluR3 vs. mGluR3-I788A, p=8.2E-10). mGluR2-P778A is not significantly different from mGluR2 (two-way ANOVA, p=0.27). (G) LY48 calcium imaging concentration response curves for mGluR2 and mGluR2-I779A (EC$_{50}$ = 5.46 ± 0.01). mGluR2-I779A has ~31% lower efficacy than mGluR2 at saturating LY48. All values are normalized to saturating (1 mM) glutamate. * indicates statistical significance (two-way ANOVA, p=6.2E-9). (H) Glutamate FRET concentration response curve for mGluR2 and mGluR2-I779A (EC$_{50}$ = 33.63 ± 0.07). All values are normalized to saturating (1 mM) glutamate. * indicates statistical significance (two-way ANOVA, p=0.00060). (I) Glutamate FRET concentration response curve for mGluR3 and mGluR3-I788A (EC$_{50}$ = 0.66 ± 0.05). All values are normalized to saturating (1 mM) glutamate. * indicates statistical significance (two-way ANOVA, p=0.030). Associated figure supplement includes *Figure 5—figure supplement 1*.

The online version of this article includes the following source data and figure supplement(s) for figure 5:

**Source data 1.** Data and statistics for Figure 5 and associated supplemental figure.
**Figure supplement 1.** Further characterization of TM6 in mGluR activation.

similar decrease in TMD dimerization (*Figure 5B*). Importantly, surface expression did not differ between wild-type and mutant constructs (*Figure 5C*; *Figure 5—figure supplement 1B*).

To determine whether functional differences exist between the mutants, we performed calcium imaging experiments with mutations in the full-length mGluR2 background. mGluR2-I779W, -I779A, and -P778A-I779A all showed right-shifted concentration response curves compared to mGluR2 while mGluR2-P778A was not significantly different but did show a modest left-shift (*Figure 5D,F*; *Figure 5—figure supplement 1D,E*). Similar to mGluR2, mGluR3-I788A showed a 19-fold rightward shift in glutamate response compared to wild-type (*Figure 5E,F*; *Figure 5—figure supplement 1F*), suggesting that this residue indeed plays an important conserved role in group II mGluR activation.

Having confirmed the importance of inter-TM6 interactions to orthosteric activation, we asked if this interface also influences allosteric drug activation. It remains unclear how similar the orthosteric and allosteric activation pathways are given that these ligands bind at very distant sites and are able to activate mGluRs independently of each other. mGluR2 and mGluR4 PAMs can activate full-length receptors even in the presence of orthosteric antagonists suggesting that LBD-driven re-arrangement can be bypassed by TMD-targeting compounds (*Gutzeit et al., 2019*; *Rovira et al., 2015*). We tested the response of mGluR2-I779A to LY48 and found a 12-fold rightward shift and a 29.9% reduction in maximal PAM efficacy compared to saturating glutamate (*Figure 5G*). Together, these results clearly indicate that TM6 interactions also play a role in allosteric activation.

Next, we asked how perturbing the TM6 interface would alter the conformational dynamics detected by our TMD FRET sensors. We introduced the I779A mutation into the mGluR2 inter-sub-unit FRET sensor and observed a modest right shift in the concentration response curve (*Figure 5H*; *Figure 5—figure supplement 1G*). While the functional assay revealed an 11-fold rightward shift for I779A, the change in EC$_{50}$ was only two-fold, suggesting that this sensor primarily senses steps in the activation process that occur prior to TM6 interface formation or conformational changes stabilized by the TM6 interface. A similar, small right-shift was observed with the mGluR3 inter-TMD FRET sensor (*Figure 5I*; *Figure 5—figure supplement 1H*). Combined, these results underscore the importance of the TM6 interface in the conformational changes that drive group II mGluR activation.

Having determined that both the cytosolic end of TM4 and the extracellular end of TM6 shape the mGluR activation process, we next sought to further clarify the specific role of each region in mediating TMD dimerization with the hypothesis that a TM4 to TM6 interface switch occurs during both orthosteric and allosteric activation. This model is both based on our data and aforementioned previous cross-linking and structural studies (*Koehl et al., 2019*; *Xue et al., 2015*). We decided to conduct SiMPull experiments on wild-type and mutant receptors in the presence of ligands to investigate effects on receptor dimerization.

We previously found that allosteric drugs did not alter the dimerization of the mGluR2 TMD (*Gutzeit et al., 2019*). However, we anticipated that mutating either the TM4 or TM6 interface (*Figure 6A*) would sensitize the mGluR2-TMD to shifts in dimerization propensity produced by favoring distinct states. Indeed, while wild-type mGluR2-TMD showed no sensitivity to PAM or NAM treatment, mGluR2-TMD-3xTM4 showed a clear increase in dimerization following treatment with a PAM (TASP) while mGluR2-TMD-I779A showed a clear increase following treatment with a NAM (MNI 137) (*Figure 6C*; *Figure 6—figure supplement 1A*). This result is consistent with a model where PAM treatment shifts the dimer toward a TM6 interface enabling rescue of reduced TM4-mediated dimerization for 3xTM4, whereas NAM treatment shifts the dimer toward a TM4 interface

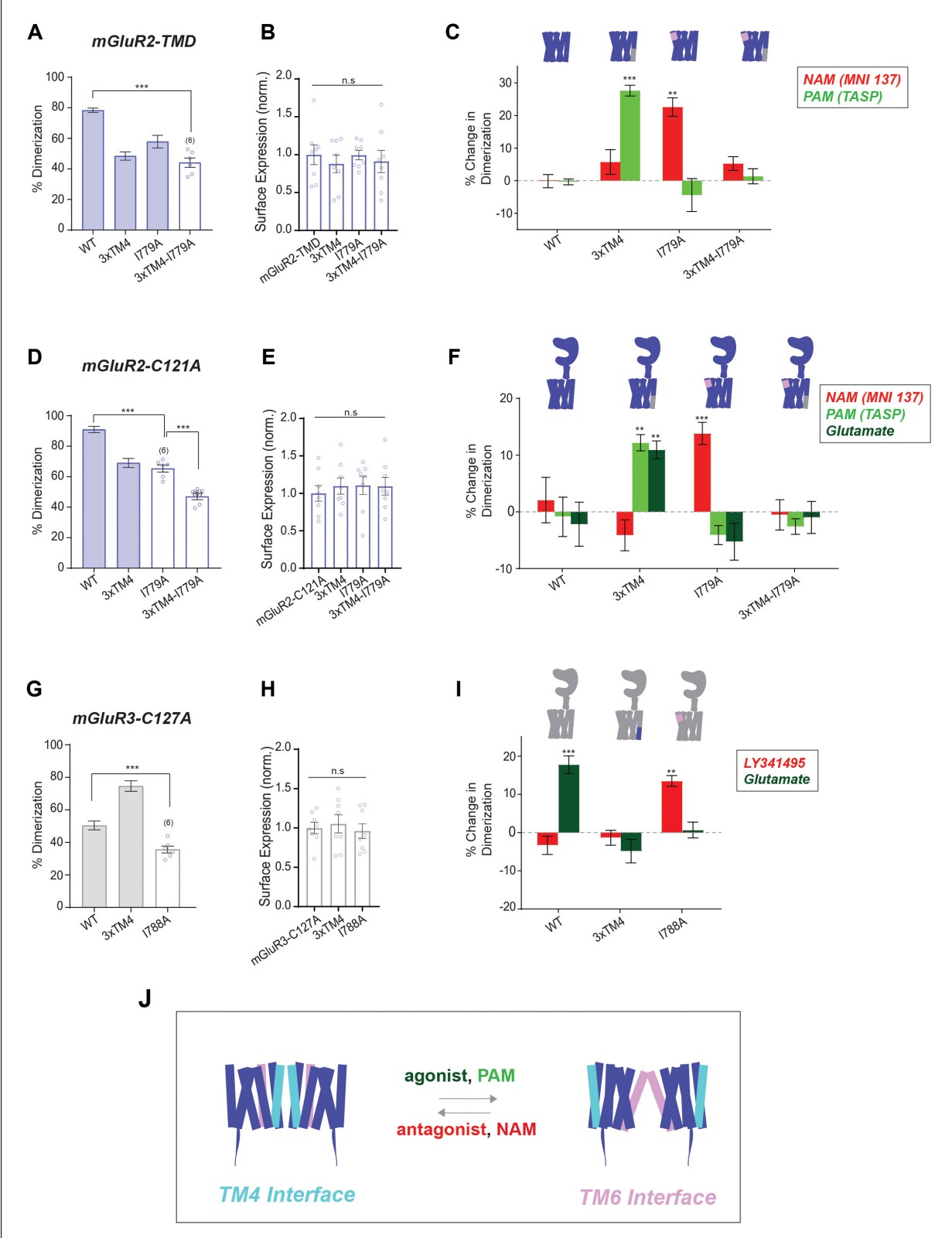

**Figure 6.** SiMPull analysis of ligand induced inter-TMD dimer rearrangement. (**A**) Bar graph showing percent dimerization for SNAP-tagged constructs labeled with LD555. * indicates statistical significance (one-way ANOVA, p=3.9E-7; Tukey-Kramer for mGluR2-TMD vs. mGluR2-TMD-3xTM4-I779A p=5.1E-7). Shaded bars indicate data is repeated from *Figures 2C* and *5B*. (**B**) Bar graph showing surface expression for constructs in (**A**). Values are normalized to SNAP-mGluR2-TMD. Expression is not significantly different between constructs (one-way ANOVA, p=0.86). (**C**) Bar graph showing the

*Figure 6 continued on next page*

Figure 6 continued

percent change in dimerization compared to no drug conditions for SNAP-tagged constructs labeled with LD555. * indicates statistical significance (one-way ANOVA, p=4.4E-16; Tukey-Kramer for mGluR2-TMD-3xTM4 vs. mGluR2-TMD-3xTM4 + TASP p=7.1E-6, for mGluR2-TMD-I779A vs. mGluR2-TMD-I779A + MNI 137 p=0.0036). (D) Bar graph showing percent dimerization for SNAP-tagged constructs labeled with LD555. * indicates statistical significance (one-way ANOVA, p=7.4E-11; Tukey-Kramer for mGluR2-C121A vs. mGluR2-C121A-I779A p=6.7E-7, for mGluR2-C121A-I779A vs. mGluR2-C121A-3xTM4-I779A p=0.00021). Shaded bars indicate data is repeated from *Figures 1F* and *2E*. (E) Bar graph showing surface expression for constructs in (D). Values are normalized to SNAP-mGluR2-C121A. Expression is not significantly different between constructs (one-way ANOVA, p=0.89). (F) Bar graph showing the percent change in dimerization compared to no drug conditions for SNAP-tagged constructs labeled with LD555. * indicates statistical significance (one-way ANOVA, p=1.4E-31; Tukey-Kramer for mGluR2-C121A-3xTM4 vs. mGluR2-C121A-3xTM4 + TASP p=0.0029, for mGluR2-C121A-3xTM4 vs. mGluR2-C121A-3xTM4 + Glu p=0.0093, for mGluR2-C121A-I779A vs. mGluR2-C121A-I779A + MNI 137 p=0.036). (G) Bar graph showing percent dimerization for SNAP-tagged constructs labeled with LD555. * indicates statistical significance (one-way ANOVA, p=1.2E-7; Tukey-Kramer for mGluR3-C127A vs. mGluR3-C127A-I788A p=0.0033). Shaded bars indicate data is repeated from *Figures 1F* and *2G*. (H) Bar graph showing surface expression for constructs in (G). Values are normalized to SNAP-mGluR3-C127A. Expression is not significantly different between constructs (one-way ANOVA, p=0.79). (I) Bar graph showing the percent change in dimerization compared to no drug conditions for SNAP-tagged constructs labeled with LD555. * indicates statistical significance (one-way ANOVA, p=1.5E-12; Tukey-Kramer for mGluR3-C127A vs. mGluR3-C127A + Glu p=9.8E-7, for mGluR3-C127A-I788A vs. mGluR3-C127A-I788A + LY34 p=0.011). The number of movies analyzed is shown in parentheses. Error bars represent s. e.m. (J) Schematic summarizing data indicating that TM4 mediates inactive interfaces which reorient to a TM6 interface upon activation by PAMs or agonists. Associated figure supplement includes *Figure 6—figure supplement 1*.

The online version of this article includes the following source data and figure supplement(s) for figure 6:

**Source data 1.** Data and statistics for Figure 6 and associated supplemental figure.

**Figure supplement 1.** Analysis of ligand induced inter-TMD rearrangement by SiMPull.

to rescue the reduced TM6-mediated dimerization of I779A. Supporting this, simultaneously mutating both interfaces with 3xTM4 and I779A prevented any PAM- or NAM-induced dimerization shifts (*Figure 6A,C*; *Figure 6—figure supplement 1A*), but did not alter surface expression levels (*Figure 6B*).

We conducted the same experiments in the mGluR2-C121A background (*Figure 6D*) to assess whether this model also described dimer rearrangement for full-length receptors. Similar to the wild-type mGluR2-TMD, mGluR2-C121A dimerization was unchanged by ligands (*Figure 6F*; *Figure 6—figure supplement 1B*). However, mGluR2-C121A-3xTM4 showed a clear increase in dimerization following treatment with a PAM or glutamate (*Figure 6F*; *Figure 6—figure supplement 1B*) and mGluR2-C121A-I779A showed increased dimerization following treatment with a NAM (*Figure 6F*; *Figure 6—figure supplement 1B*). Lastly, mGluR2-C121A-3xTM4-I779A retained reduced dimerization (47.0 ± 1.9%) with and without all ligands (*Figure 6D,F*; *Figure 6—figure supplement 1B*) while showing normal surface expression (*Figure 6E*).

Next, we wondered whether the same inactive and active interfaces could be observed for mGluR3. Since mGluR3-C127A already shows reduced dimerization compared to mGluR3 (*Figure 6G*), we anticipated that the addition of ligands might alter dimerization without further mutation to the TMD. Indeed, addition of LY34 decreased dimerization while addition of glutamate increased dimerization (*Figure 6I*; *Figure 6—figure supplement 1C*). Increasing dimerization propensity at the TM4 interface in mGluR3-C127A-3xTM4 prevented the ligand-induced effects on dimerization (*Figure 6I*; *Figure 6—figure supplement 1C*), likely due to a relative balance in dimerization propensity between TM4 and TM6 in this construct as is seen with mGluR2. Finally, we tested mGluR3-C127A-I788A and observed a small increase in dimerization following the addition of LY34 and no difference in the presence of glutamate (*Figure 6I*; *Figure 6—figure supplement 1C*) and unaltered surface expression (*Figure 6H*). Overall, these results establish the combination of drug treatment with single molecule pulldown as a means to assess ligand-induced membrane protein assembly and support a model of ligand-regulated shift between TM4 and TM6-containing dimer interfaces (*Figure 6J*).

## Inter-TMD interactions fine-tune the assembly of mGluR2/3 heterodimers

An increasingly appreciated aspect of mGluR molecular diversity is the formation of heterodimers which have been shown to have unique pharmacological and functional properties (*Levitz et al., 2016*; *Habrian et al., 2019*; *Yin et al., 2014*; *Moreno Delgado et al., 2017*; *Liu et al., 2017*;

*Sevastyanova and Kammermeier, 2014*; *Werthmann et al., 2020*). We recently showed that extensive co-expression of mGluR subtypes occurs throughout the brain and that a complex set of preferences determines the relative propensities for heterodimerization of different mGluR pairs (*Lee et al., 2020*). Among the most preferred pairs are mGluR2 and mGluR3 which readily form heterodimers following heterologous expression and can be co-immunoprecipitated from the frontal cortex. Given our finding that mGluR2 and mGluR3 homodimers have distinct inter-TMD interaction properties, we asked how this would manifest in the mGluR2/3 heterodimer: Are mGluR2 and mGluR3 TMDs able to interact? Does the TMD-TMD interface resemble mGluR2 or mGluR3 homodimers?

We first performed two-color SiMPull experiments (Materials and methods) to determine if the isolated HA-SNAP-mGluR2-TMD construct can pull-down CLIP-mGluR2-TMD or CLIP-mGluR3-TMD (*Figure 7A*). We labeled HA-SNAP-mGluR2-TMD with BG-LD655 (red) and either CLIP construct with BC-DY547 (green) and calibrated the ratio of the surface expression of the SNAP and CLIP-tagged constructs to be comparable between conditions (*Figure 7—figure supplement 1A,B*), before isolating receptor complexes using the anti-HA antibody (*Figure 7A*) as was done for previous one-color experiments. While SNAP-mGluR2-TMD was able to pull-down both constructs, a higher level was observed for CLIP-mGluR2-TMD (*Figure 7B,C*), suggesting that TMD heterodimers can form but the TMD dimerization propensity of mGluR2 homodimers is higher than that of mGluR2/3 heterodimers. We performed the reverse experiment with HA-SNAP-mGluR3-TMD (*Figure 7D*) and found higher levels of pulldown for CLIP-mGluR2-TMD compared to CLIP-mGluR3-TMD (*Figure 7E,F*), suggesting that the inter-TMD assembly of mGluR3 homodimers has a slightly lower propensity than that of mGluR2/3 heterodimers.

We next investigated inter-TMD interaction of mGluR2/3 homodimers in the full-length context. Consistent with previous work (*Lee et al., 2020*), at comparable ratios of surface-expression, full-length mGluR2 preferentially assembles with mGluR3 (*Figure 7G–I*; *Figure 7—figure supplement 2A–C*). However, mutating the intersubunit disulfide-forming cysteine (C121A in mGluR2; C127A in mGluR3) resulted in a dramatic swap of the dimerization preference. HA-SNAP-mGluR2-C121A pulled down CLIP-mGluR2-C121A at nearly wild-type levels, consistent with the minor effect of this mutation on mGluR2 homodimerization (*Figure 1F*) but pulled down a much lower level of CLIP-mGluR3-C127A (*Figure 7H–I*). Consistent with isolated TMD results, SNAP-mGluR3-C127A also showed a weak preference for CLIP-mGluR2-C121A over CLIP-mGluR3-C127A (*Figure 7J–K*; *Figure 7—figure supplement 2D–F*). To test whether inter-TM4 interactions contribute to the distinct assembly properties of mGluR2/3 heterodimers, we introduced the 3xTM4 mutation into CLIP-mGluR3-C127A and observed enhanced pulldown via SNAP-mGluR2-C121A (*Figure 7H–I*). While this construct was unable to rescue mGluR2/3 heterodimerization back to the level of wild-type receptors, this result clearly indicates that this stretch of TM4 is a determinant of the assembly of mGluR2/3 heterodimers. Similarly, introducing the 3xTM4 mutation into CLIP-mGluR2-C121A decreased the efficiency of pulldown via SNAP-mGluR3-C127A (*Figure 7J,K*). Together these data indicate that mGluR2/3 TMD dimers show an intermediate propensity compared to their parent homodimers (*Figure 7L*), further supporting the TMD dimer interface as a site of structural and functional diversity between mGluR subtypes.

## Discussion

Despite advancements in our understanding of dimeric family C GPCR assembly and activation at the level of the extracellular domains (*Ellaithy et al., 2020*), it has remained less clear how TMDs interact and rearrange in this process. Our findings provide new insight into the molecular basis of mGluR TMD association and argue for a perspective that takes into account an ensemble of inter-TMD dimerization states that are dynamically populated with distinct occupancies across subtypes.

Our finding that swapping TMDs between mGluR2 and mGluR3 swaps dimerization propensity demonstrates that subtype-specific interactions between TMDs affect full-length mGluR dimer assembly. Furthermore, our discovery that three membrane-facing residues at the cytoplasmic end of TM4 confer differences between mGluR2 and mGluR3 builds on previous reports of TM4 as a putative mGluR2 TMD interface based on cross-linking (*Xue et al., 2015*; *Koehl et al., 2019*; *González-Maeso et al., 2008*), while adding a critical new perspective that dimer assembly can vary dramatically within a receptor subfamily and even between two closely related receptors with >70%

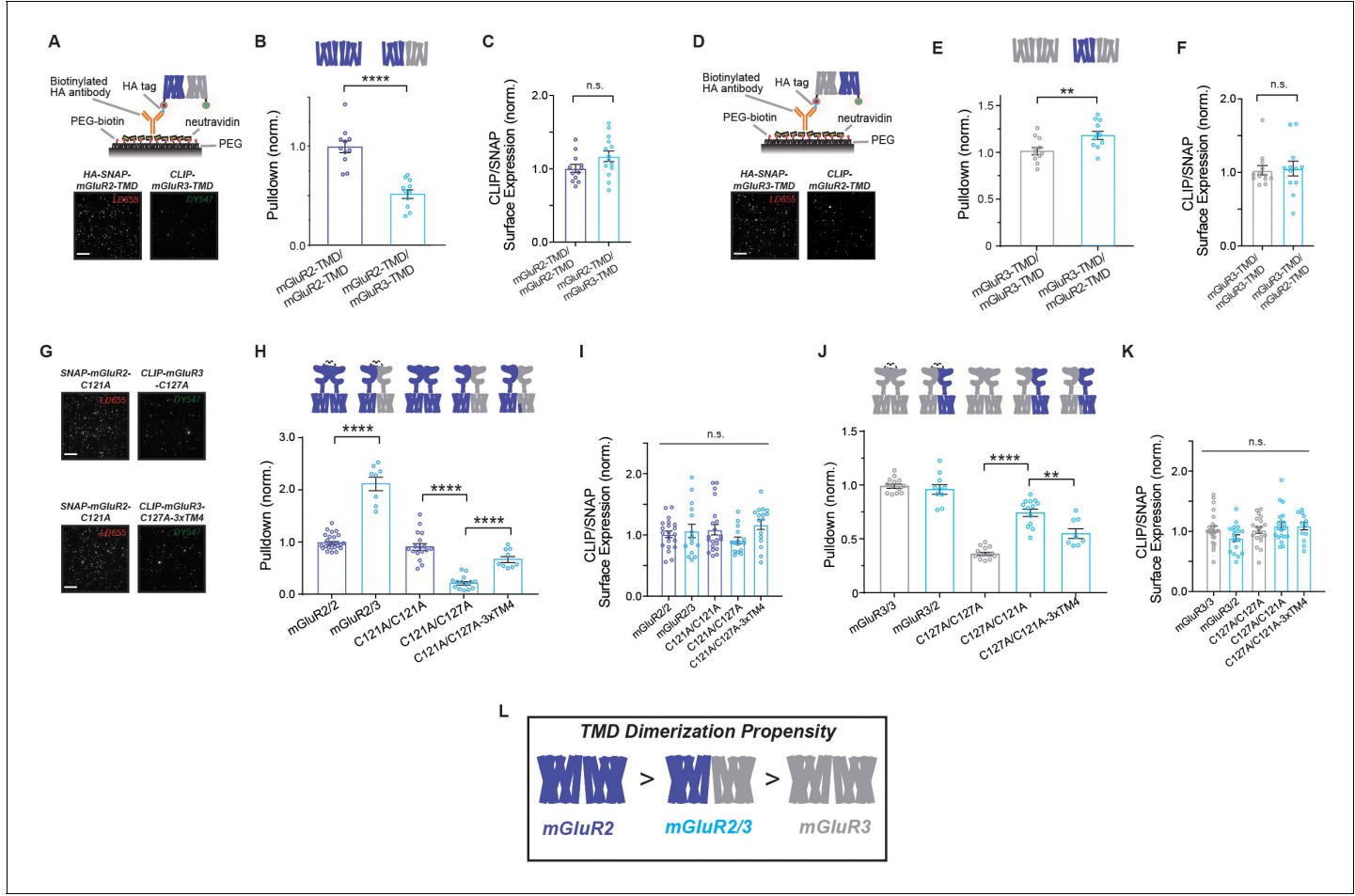

**Figure 7.** mGluR2/3 heterodimers show intermediate inter-TMD dimerization propensity compared to parent homodimers. (**A**) Top, schematic of two-color heterodimer SiMPull experiments where HA-SNAP-mGluR2-TMD (labeled with LD655) is able to immobilize CLIP-mGluR3-TMD (labeled with DY-547). Bottom, representative image showing efficient pulldown in both channels, indicative of heterodimerization of mGluR2 and mGluR3 TMDs. (**B**) Summary bar graph showing the efficiency of pulldown of either CLIP-mGluR2-TMD or CLIP-mGluR3-TMD by SNAP-mGluR2-TMD. * indicates statistical significance (unpaired t-test; p=2.2E-6). (**C**) Bar graph showing ratio of surface expression of CLIP- to SNAP-tagged constructs in (**B**). Values are normalized to SNAP-mGluR2-TMD + CLIP-mGluR2-TMD. Expression ratio is not significantly different between constructs (unpaired t-test, p=0.087). (**D**) Top, schematic of two-color heterodimer SiMPull experiments where HA-SNAP-mGluR3-TMD (labeled with LD655) is able to immobilize CLIP-mGluR2-TMD (labeled with DY-547) via an anti-HA antibody. Bottom, representative image showing efficient pulldown in both channels, indicative of heterodimerization. (**E**) Summary bar graph showing the efficiency of pulldown of either CLIP-mGluR2-TMD or CLIP-mGluR3-TMD by SNAP-mGluR2-TMD. * indicates statistical significance (unpaired t-test; p=0.0097). (**F**) Bar graph showing ratio of surface expression of CLIP- to SNAP-tagged constructs in (**E**). Values are normalized to SNAP-mGluR3-TMD + CLIP-mGluR3-TMD. Expression ratio is not significantly different between constructs (unpaired t-test, p=0.84). (**G**) Representative images showing SiMPull of full-length CLIP-mGluR3 by full-length HA-SNAP-mGluR2. (**H**) Summary bar graph showing efficiency of pulldown of wild type and mutant full-length CLIP-tagged mGluR2 and mGluR3 by HA-SNAP-mGluR2. * indicates statistical significance (one-way ANOVA, p=1.5E-28; Tukey-Kramer for mGluR2/mGluR2 vs. mGluR2/mGluR3, p=1E-14; for C121A/C121A vs. C121A/C127A, p=1E-14; for C121A/C127A vs. C121A/C127A-3xTM4, p=3.7E-6). (**I**) Bar graph showing ratio of surface expression of CLIP- to SNAP-tagged constructs in (**H**). Values are normalized to SNAP-mGluR2 +CLIP-mGluR2. Expression ratio is not significantly different between constructs (one-way ANOVA, p=0.26). (**J**) Summary bar graph showing efficiency of pulldown of wild type and mutant full-length CLIP-tagged mGluR2 and mGluR3 by HA-SNAP-mGluR3. * indicates statistical significance (one-way ANOVA, p=2.3E-21; for C127A/C127A vs. C127A/C121A, p=8E-12; for C127A/C121A vs. C127A/C121A-3xTM4, p=0.0017). (**K**) Bar graph showing ratio of surface expression of CLIP- to SNAP-tagged constructs in (**H**). Values are normalized to SNAP-mGluR3 +CLIP-mGluR3. Expression ratio is not significantly different between constructs (one-way ANOVA, p=0.11). (**L**) Summary schematic showing relative dimerization propensities of mGluR2, mGluR2/3 and mGluR3 TMD combinations. Associated figure supplements include **Figure 7—figure supplements 1–2**.

The online version of this article includes the following source data and figure supplement(s) for figure 7:

**Source data 1.** Data and statistics for Figure 7 and associated supplemental figures.
**Figure supplement 1.** Further analysis of homo- and hetero-dimerization propensities of isolated group II mGluR TMDs.
**Figure supplement 2.** Further analysis of homo- and hetero-dimerization propensities of isolated full-length group II mGluRs.

sequence identity. The conserved triple alanine motif in TM4 of mGluR2 may support an increased TMD dimerization propensity, as alanine displays a preference to be buried when in an alpha-helical transmembrane helix (*Ulmschneider and Sansom, 2001*) and can contribute to tight helix-helix packing due to its short side chain (*Senes et al., 2000*; *Eilers et al., 2000*). Conversely, the mGluR3 TM4-containing dimer region includes serine, phenylalanine, and glycine residues. Consistent with the decreased TMD dimerization of mGluR3, the hydrophobic and bulky nature of phenylalanine is less favorable for close helix-helix packing in membrane proteins and serine contributes to trans-membrane packing less favorably than alanine (*Eilers et al., 2000*). Glycine often occurs at helix-helix interfaces in transmembrane regions (*Senes et al., 2000*) but experiments and molecular dynamics simulations have argued that glycine can be helix destabilizing in membrane proteins (*Dong et al., 2012*; *Högel et al., 2018*).

Our data on mGluR2/3 heterodimer assembly (*Figure 7*) confirms the dominant role of inter-LBD interfaces in determining relative dimer propensities (*Levitz et al., 2016*; *Lee et al., 2020*). However, the dramatic effect of disruption of inter-subunit disulfide bonds reveals an unexpected critical role for this covalent linkage in determining the enhanced assembly of mGluR2/3 heterodimers compared to mGluR2 homodimers. TM4 is a clear contributor to the assembly of mGluR2/3 heterodimers with increased dimerization observed for heterodimers compared to mGluR3 homodimers and reduced dimerization compared to mGluR2 homodimers. This likely shapes the activation properties of this complex which we have previously shown to have intermediate glutamate affinity, basal activity, and cooperativity compared to parent homodimers (*Levitz et al., 2016*). The considerable variability observed within TM4 and other helices across group I and group III mGluRs (*Figure 2— figure supplement 1*) suggests that further complexity likely exists in TMD dimerization beyond the group II subtypes. We have previously shown that mGluR2 TMD has a greater dimerization propensity than mGluR1 and mGluR5 (*Gutzeit et al., 2019*) which is in line with the lack of an inter-TMD interface in the inactive, antagonist-bound mGluR5 cryo-EM structure (*Koehl et al., 2019*). Consistent with this, only mGluR2 shows the triple alanine motif at the bottom half of TM4 suggesting that while TM4 interfaces may contribute to dimerization across all subtypes, mGluR2 maintains unique interactions between TMDs.

Our study also provides insight into the role of inter-TMD interaction in the activation of mGluRs. While increasing the dimerization propensity via the TM4 interface hinders receptor activation and inter-TMD re-arrangement, decreasing the dimerization propensity via the TM4 interface has more complex effects that can either enhance activation, in the case of 3xTM4 on mGluR2, or impair activation, in the case of harsher tryptophan mutants (*Figures 3* and *4*). Based on these results we anticipate that outward-facing residues in TM4 can both support inactive states, as seen in wild-type mGluR2 and a previous study where TM4/TM5 crosslinking impaired receptor activation (*Xue et al., 2015*), but can also contribute to the receptor activation pathway. Such observations are in-line with the MSM analysis of our CG MD simulations, which reveals many distinct modes of TM4-based dimerization, including those that engage helices other than TM4, and especially TM1. It is possible that distinct modes are associated with inhibited inactive states while others are likely to be intermediates along an activation pathway that converges on TM6 interfaces. Notably, the MSM analysis of mGluR2 reveals a direct reactive flux between the identified macrostate of TM4-containing dimers and the macrostate containing TM6 helices, whereas equivalent macrostates identified for mGluR3 are only indirectly connected. Most clearly, our ligand-coupled SiMPull analysis (*Figure 6*), strongly supports a TM4 to TM6 rearrangement along the activation pathway with both orthosteric and allosteric agonists favoring a TM6 interface while antagonists and negative allosteric modulators favor a TM4 interface. It is notable that TM6-containing interfaces are formed with much lower probability with respect to others in our inactive-state MD simulations, suggesting that an active TMD and/or active LBDs are required to engage this interface. Finally, TM1 interfaces are prominent in our MD simulations, with TM1-containing dimers being among the most probable configurations alongside TM4-containing dimers. While our computations suggest that TM1 is involved in a variety of kinetically interchangeable asymmetric dimers, TM1 interfaces could also be relevant for higher order oligomers, as proposed for mGluR2 in neurons (*Møller et al., 2018*). Future work will be needed to assess the contribution of TM1, and other helices not experimentally probed here but identified as possible interfacial participants (e.g. TM5 for mGluR2 and TM7 for mGluR3), to mGluR dimerization. Importantly, the SiMPull experiments employed here allowed us to obtain 'snapshots' of dimerization propensities in a relevant cellular context, but they may not necessarily reflect thermodynamic

equilibrium since metastable states might become kinetically trapped. Thus, future in vitro work at defined concentrations is required in order to gain a complete biophysical understanding of mGluR dimerization.

Together our data, in addition to supporting a TM4 to TM6 activation-associated transition, raises the question of whether inter-TMD monomer-dimer transitions are a critical aspect of mGluR activation. While our MSM analysis suggests that many interfaces likely contribute to the activation pathway, we hypothesize that TMDs transition between a monomeric assembly, TM4-containing dimer assemblies and a TM6-containing active dimer interface that may be stabilized by agonists, PAMs, and G proteins. Consistent with this idea, it is likely that the intersubunit FRET sensors reported here are able to sense a mix of monomer-dimer transition and dimer rearrangement. Based on the fast transitions previously observed with similar inter-TMD FRET sensors on the same ~1–10 ms timescale as inter-LBD rearrangement (*Grushevskyi et al., 2019*; *Hlavackova et al., 2012*; *Olofsson et al., 2014*), a TMD monomer-dimer transition would likely be driven by the agonist-induced relaxed to active LBD transition where the intermediate cysteine-rich domains (CRDs) come into closer proximity (*Koehl et al., 2019*; *Huang et al., 2011*; *Liauw et al., 2021*). Given the partial sensitivity of the intersubunit FRET sensors to NAMs and PAMs, it is reasonable to posit that allosteric drugs primarily control the reorientation between TM4 and TM6 interfaces but not the transition between TMD monomers and dimers. This is in line with our previous mGluR2 inter-TMD FRET measurements with isolated domains where changes in FRET but not monomer: dimer equilibrium were observed with PAMs and NAMs (*Gutzeit et al., 2019*). The clear effects of TM4 and TM6 interface mutations on PAM responses in our study (*Figure 3*, *Figure 6*) are consistent with this. However, PAMs can activate isolated monomeric mGluR2 (*El Moustaine et al., 2012*) and a previous study showed only modest inhibition of PAM responses in a TM4/TM5-crosslinked mGluR (*Xue et al., 2015*), raising the possibility that PAMs can alter intra-TMD conformation independently of inter-TMD rearrangement.

Further structural studies are critical to obtain a deeper understanding of mGluR conformational changes, especially at the TMD interface. The proposed metastability of TM4 interfaces may explain why they have not been observed in static structures solved by crystallography or cryo-EM, although our steering simulations suggest that symmetric TM4-containing dimers should be relatively stable in the full-length mGluR2. In addition, further computational studies, especially with all-atom and full-length receptor models, and spectroscopic measurements, especially at the single-molecule level, will be needed to decipher the complex orthosteric and allosteric activation dynamics of different homo- and hetero-dimeric mGluR subtypes. Of critical importance is understanding how the molecular motions of the LBDs and CRDs on the millisecond and sub-millisecond timescales are transmitted to the many micro- and macro-states of the TMDs.

What is the biological relevance of the modulatory effects of TMD dimerization differences between subtypes? Variable inactive-state TMD dimerization likely contributes to the tuning of the mGluR activation dynamics to the synaptic roles of each particular subtype. The 3- to 20-fold shifts in apparent glutamate affinity observed when swapping residues between mGluR2 and mGluR3, while subtle, are highly relevant in the context of the spatiotemporally complex dynamics of synaptic and extra-synaptic glutamate (*Reiner and Levitz, 2018*). Our data also suggest that in addition to controlling apparent glutamate sensitivity, relative inter-TMD dimerization propensity tunes the basal activity, and likely also shapes the signaling kinetics of different mGluRs. Outside of the mGluR homo- and heterodimer context, the proposed inactive TM4 interface is intriguing to consider in the context of the mGluR2-5HT$_{2A}$R interaction, which was reported to require the same three alanine residues for heterocomplex formation (*Moreno et al., 2012*; *Moreno et al., 2016*). Our data suggests that formation of this heterocomplex would either require activation of mGluR2 to induce TMD reorientation away from a TM4 mGluR interface or stabilization of a state where TMD dimers are unable to interact. In line with this hypothesis, previous work found that co-expression of mGluR2 and 5-HT$_{2A}$R increased glutamate-elicited G$_{i/o}$ activation (*Fribourg et al., 2011*). More work will be required to fully understand this unique inter-family interaction and if such inter-TMD interactions occur beyond the prototypical mGluR2-5HT$_{2A}$R complex.

Overall, this study has revealed that closely related GPCRs can have distinct inter-subunit interactions that drive differences in assembly, conformational dynamics and signaling. Given the high degree of homology between mGluR2 and mGluR3, this suggests that subtle differences in

quaternary structure may be a major contributor to the unique functional properties of distinct GPCR subtypes within the same family.

## Materials and methods

### Cell culture, molecular cloning, and gene expression

HEK 293 T cells were purchased from ATCC and validated by DNA profiling (Bio-Synthesis, Inc). They tested negative for mycoplasma using a commercial kit. The cells were cultured in DMEM with 5% FBS on poly-L-lysine-coated glass coverslips. Lipofectamine 2000 (Thermo Fisher) was used for transfection of DNA plasmids. For single-molecule pulldown (SiMPull) experiments and surface expression measurements, cells were transfected with 0.7 μg DNA of HA- and SNAP-tagged receptor constructs. For calcium imaging experiments, cells were transfected with the mGluR of interest, a $G\alpha_{qi3}$ chimera (*Conklin et al., 1993*) and GCaMP6f, in a 7:5:3 ratio with 0.7 μg of receptor. For FRET experiments, cells were transfected with either 0.7 μg SNAP-tagged constructs or 0.7 μg Halo-Tag constructs. For inter-TMD FRET experiments, HaloTags were inserted in ICL2 at an identical position in both mGluR2 and mGluR3 (directly following I669 in mGluR3 and I660 in mGluR2). An alanine-serine linker precedes the HaloTag and a leucine-glutamatic acid linker follows it. SNAP-mGluR-TMD clones were made as described in *Gutzeit et al., 2019*. All mutations were made using standard PCR-based techniques. TMD swap chimeras were made using Gibson assembly (*Gibson et al., 2009*). For mGluR3-mGluR2TMD, Q558 to P821 of mGluR2 was inserted into mGluR3 between P556 and Q831. For mGluR2-mGluR3TMD, E567 to P830 of mGluR3 was inserted into mGluR2 between P835 and Q800.

### Surface expression measurements

Receptor constructs were expressed in HEK 293 T cells for 24–48 hr prior to labeling with 1 μM SNAP-Surface Alexa Fluor 546 (NEB) in extracellular solution at 37°C for 45 min. For two-color experiments with co-expression of SNAP- and CLIP-tagged receptors, cells were labeled with 1 μM BG-LD655 followed by 2 μM BC-DY547 (NEB). After labeling, cells were washed in extracellular solution to remove excess dye. Next, cells were imaged with a 60x objective on an inverted microscope (Olympus IX73). ImageJ was used to analyze fluorescence intensity of cell clusters. Multiple independent experiments were performed for each construct tested, including at least two separate transfections. Fluorescence intensity values for each construct were normalized to wild-type values for comparison.

### Single-molecule pulldown and subunit counting

Single-molecule pulldown (SiMPull) was performed as previously described (*Levitz et al., 2016*). Briefly, flow chambers were prepared with mPEG-passivated glass slides and coverslips doped with ~1% biotinylated mPEG to prevent non-specific sticking of proteins. Preceding each experiment, chambers were incubated with 0.2 mg/mL NeutrAvidin (ThermoFisher) for 2 min followed by 10 nM of a biotinylated anti-HA antibody (abcam ab26228) for 20–30 min. Following each conjugation step, chambers were washed with T50 buffer (50 mM NaCl, 10 mM Tris, pH = 7.5).

24–48 hr after transfection of HEK 293 T cells, the cells were labeled with 1.5 μM benzylguanine (BG)-LD555 (*Gutzeit et al., 2019*; *Qinsi et al., 2017*) or SNAP-Surface Alexa Fluor 546 (New England BioLabs) in extracellular solution at 37°C for 45 min. For two-color experiments with co-expression of SNAP- and CLIP-tagged receptors, cells were labeled with 1 μM BG-LD655 followed by 2 μM BC-DY547 (NEB). After labeling, cells were washed in extracellular solution to remove excess dye. Next, cells were transferred to $Ca^{2+}$-free PBS for 20 min to dissociate cells from coverslips. Cells were then completely removed from coverslips by gentle pipetting. Centrifugation was used to pellet cells (16,000 g for 1 min) before lysis in buffer containing 10 mM Tris, 150 mM NaCl, 1 mM EDTA, protease inhibitor cocktail, and 1.2% IGEPAL (Sigma), pH = 8.0. After 1 hr of lysis at 4°C, cells were centrifuged at 16,000 g for 20 min. Next, the supernatant was collected and kept on ice. During experiments, the cell lysate was diluted in a dilution buffer (0.1% IGEPAL, 1 mM Tris pH 8, 100 uM EDTA, 135 mM NaCl, 5 mM KCl, 9 mM HEPES, 2 mM $CaCl_2$, 1 mM $MgCl_2$) to achieve sparse immobilization of fluorophore-labeled protein on the passivated slide. Protein was added to the slide until an optimal density was reached and then the flow chamber was washed with dilution

buffer to remove unbound protein. For experiments with ligands, drugs were added to cells after dye labeling (during dissociation step in $Ca^{2+}$-free PBS) and maintained throughout the entire experiment at saturating concentration.

Single-molecule imaging was conducted with a 100x objective on an inverted microscope (Olympus IX83) in total internal reflection fluorescence (TIRF) mode. Movies were recorded with a scMOS camera (Hamamatsu ORCA-Flash4v3.0) at 20 Hz with 50 ms exposure. The fluorophores were excited with a 561 nm laser. Multiple independent experiments were performed for each condition tested, including at least two separate transfections and protein preparations. LabVIEW was used for data analysis as previously described (*Ulbrich and Isacoff, 2007*).

Single-molecule fluorescence time traces were manually classified as having 1, 2, 3, or 4 bleaching steps or were discarded if no clear bleaching steps were identified. The photobleaching count distribution is then used to determine the percentage of two-step photobleaching:

$$\%2-step\,photobleaching = \frac{number\,of\,spots\,bleaching\,in\,2\,steps}{total\,analyzed\,spots} * 100$$

To calculate percent dimerization, the two-step photobleaching values were min-max normalized. Here, the min value corresponds to fluorescence background (5%) and the max value corresponds to % two-step photobleaching observed for full-length mGluR2, an obligate dimer (55%). This calculation makes no correction for expression level.

$$\%dimerization = \frac{((Observed\,value) - Min\,value) * 100}{Max\,value - Min\,value}$$

For two-color SiMPull experiments, fluorescence emission was separated using a 635 nm long pass dichroic filter place in front of separate scMOS cameras. Two-color SiMPull movies were recorded first by exciting LD655 with 640 nm laser at 20 Hz with 50 ms exposure until all spots bleached. Then, DY547 fluorescent spots were excited with 561 nm at the same imaging speed and exposure time. For analysis, the total number of spots in each channel was counted. The number of non-specific background spots was determined from control pulldowns of CLIP-tagged receptor only and was subtracted from the number of DY547 spots before calculating a ratio of DY547 to LD655 spots. These ratios were then normalized to the ratio for the corresponding homodimer condition (i.e. SNAP-mGluR2 +CLIP-mGluR2 or SNAP-mGluR3 +CLIP-mGluR3) from the same day. At least two separate experimental days were performed then averaged to produce bar graphs.

Sequence Conservation Analysis mGluR amino acid sequences were obtained from UniProt and aligned using the Multiple Sequence Alignment tool from Clustal Omega. Conservation is defined by the chemical characteristics of the amino acid side chain (R group): aliphatic (glycine, alanine, valine, leucine, isoleucine); hydroxyl or sulfur-containing (serine, cystine, threonine, methionine); cyclic (proline); aromatic (phenylalanine, tyrosine, tryptophan); basic (histidine, lysine, arginine); acidic and their amides (aspartate, glutamate, asparagine, glutamine).

## Homology modeling

An initial model of the transmembrane domain of inactive rat mGluR2 was obtained by homology modeling using the MODELLER 9.19 software (*Sali and Blundell, 1993*) and the 2.2 Å X-ray crystal structure of inactive mGluR5 (PDB ID: 6FFI *Christopher et al., 2019*) as a structural template. Missing atomic coordinates of extracellular loop 2 (EC2; residues 707–710 of rat mGluR2) were taken from PDB ID 6FFH (*Christopher et al., 2019*) whereas missing atomic coordinates of intracellular loop 2 (IC2; residues 669–674 of rat mGluR2) were generated ab initio and selected according to the lowest MODELLER DOPE score (*Shen and Sali, 2006*). To minimize differences in the modeled IC2 region between mGlu2 and mGluR3, an inactive rat mGluR3 homology model was obtained using the aforementioned mGluR2 model as a structural template and the Prime 5.9 software (*Jacobson et al., 2002*; *Jacobson et al., 2004*).

## Coarse-grained molecular dynamics simulations

Coarse-grained (CG) representations of mGluR2 and mGluR3 inactive models were obtained using the *martinize.py* script within the MARTINI 2.2 force field (*de Jong et al., 2013*). To preserve the tertiary structure of the two receptors, an elastic network was applied to pairs of backbone beads

whenever the distance between the pairs was smaller than 0.9 nm. A larger force constant of 1000 kJ mol$^{-1}$ nm$^{-2}$ was applied to pairs of beads within helical secondary structures whereas loop regions were kept more flexible with a force constant of 250 kJ mol$^{-1}$ nm$^{-2}$. Two CG copies of the same receptor were placed in a 15 × 15 × 11 nm$^3$ simulation box at a distance of 4.5 or 5.5 nm from each other. Each protomer was then rotated randomly around the axis perpendicular to the membrane plane (z-axis) to generate multiple starting configurations of neighboring protomers. Each pair of protomers was then embedded in a POPC membrane (~635 lipids) and solvated with water molecules and NaCl (0.15 M plus neutralizing ions) using the INSert membrANE (*insane*) method (*Wassenaar et al., 2015*). In total, 90 systems were generated for mGluR2 and 92 for mGluR3.

Each system was energy minimized and then equilibrated in five steps. First, a 400 ns MD simulation was run using position restraints on the backbone beads of the receptor with a force constant of 1000 kJ mol$^{-1}$ nm$^{-2}$. Four additional equilibration steps followed, each of which ran for 60 ns and with position restraints applied with decreasing force constants of 500, 100, 50, and 10 kJ mol$^{-1}$ nm$^{-2}$, respectively. These MD equilibration simulations used the velocity-rescale thermostat (*Bussi et al., 2007*) and the Berendsen algorithm (*Eslami et al., 2008*) to maintain constant temperature at 310 K and pressure at 1 bar, respectively. Production MD runs of 3 or 6 μs (depending on the shorter or longer distance between protomers in the initial configuration) for each system, amounting to 450 μs of simulation time for mGluR2 and 462 μs for mGluR3, were then carried out in the NPT ensemble at 310K and 1 bar using the velocity-rescale thermostat (*Bussi et al., 2007*) and the Parrinello-Rahman pressure coupling scheme (*Parrinello and Rahman, 1981*) respectively and using an integration timestep of 30 fs. The Coulombic and van der Waals interactions both were set to decay to zero between 0 and 1.2 nm (Coulombic) and 0.9 and 1.2 nm (van der Waals). All simulations were performed using the GROMACS 2020.4 software package.

## Markov state model analysis

The relative position of two protomers of mGluR2 or mGluR3 was characterized by calculating the number of residue contacts formed by each helix and loop on one protomer with the other protomer. Contacts were defined by a cutoff of 10 Å on the minimal distance over the beads of the residue. Each simulation frame was then classified using a label that contained the domain names of each protomer that formed more than 20 contacts with the other protomer. This classification yielded ~2400 and ~1700 microstates for the mGluR2 and mGluR3 simulations, respectively. Trajectory frames thus clustered were used to estimate a MSM with a lag time of $\tau = 20\text{ns}$, identified after inspecting the change of the implied timescales obtained from transition matrix eigenvalues for increasing lag times. The resulting transition matrix was used in the PCCA+ algorithm (*Röblitz and Weber, 2013*) to aggregate the microstates into a number of putative biologically relevant macrostates identified by a gap in the characteristic relaxation timescales. Macrostate equilibrium probabilities $\pi_I = \sum_{i \in I} \pi_i$ were calculated summing the probabilities of the corresponding microstates. To label each macrostate, we calculated the probability that each helix be at the interface by summing the conditional probabilities that a helix $D$ is present for protomer $\alpha$ in the label $\mathfrak{l}_i^{(\alpha)}$ of a microstate $i$ in the macrostate $I$:

$$p(D, \alpha, I) = \sum_{i \in I, D \in \mathfrak{l}_i^{(\alpha)}} \frac{\pi_i}{\pi_I}$$

Helices with a probability of at least 40% to be at the interface were reported in the macrostate label. Coarse-grained reactive flux between pairs of macrostates was calculated using Transition Path Theory, aggregating asymmetric macrostates that correspond to the same interface but with swapped protomers. Microstates were aggregated if (a) their labels $\mathfrak{l}_i^{(\alpha)}$ contained the same helices, but different loops, and (b) their labels were equivalent after swapping protomers, that is, $\mathfrak{l}_i^{(\alpha)} = \mathfrak{l}_j^{(\beta)}$ and $\mathfrak{l}_i^{(\beta)} = \mathfrak{l}_j^{(\alpha)}$. Macrostate contact matrices between domain $D$ on the first protomer and $D'$ on the second protomer $\langle C_{DD'} \rangle_I = \sum_i \rho_i(I) \langle C_{DD'} \rangle_i$ were calculated by averaging the microstate contacts using the PCCA+ cluster memberships $\rho$. Contact probabilities between each microstate were defined as averages over the microstate frames using a cutoff of 0.9 nm. Fitting of the Markov Models was carried out using the pyEmma library, version 2.5.7 (*Scherer et al., 2015*).

## Steered molecular dynamics simulations

A model of the dimeric full-length inactive rat mGluR2 was generated by homology modeling using the MODELLER 9.19 software (*Sali and Blundell, 1993*). The 4 Å electron microscopy structure of inactive mGluR5 (PDB ID: 6N52 *Christopher et al., 2019*) was used as a template for the extracellular domain (ECD) and extracellular loop 2. The rest of the TMD was modeled based on the 2.2 Å X-ray crystal structure of inactive mGluR5 (PDB ID: 6FFI *Christopher et al., 2019*) or generated ab initio (IC2), similarly to what was done for the TMD models. Just as for the aforementioned TMD systems, a homology model of inactive rat mGluR3 was obtained using the dimeric full-length mGluR2 model as a structural template and the Prime 5.9 software (*Jacobson et al., 2002*; *Jacobson et al., 2004*).

As reported for the TMD-only models, CG representations of mGluR2 and mGluR3 full-length models were obtained with the *martinize.py* script within the MARTINI 2.2 force field (*de Jong et al., 2013*) and the same criteria for the elastic network was enforced to preserve the tertiary structure of the receptors. The full-length dimers were placed in a $1000 \times 1000 \times 1000 \text{ nm}^3$ simulation box, respectively, and minimized in gas phase with a pseudo-PBC approach (*Konermann et al., 2018*). Briefly, non-bonded interactions were calculated using periodic boundary conditions and the Verlet neighbor search scheme, with the Coulomb and Lennard-Jones cutoffs set to 333.3 nm. To preserve the ECD dimeric interface configuration and its orientation relative to the TMD, additional positional restraints were imposed on the LBD ($1000 \text{ kJ mol}^{-1} \text{ nm}^{-2}$ on the x-, y-, and z-coordinates), the CRD ($100 \text{ kJ mol}^{-1} \text{ nm}^{-2}$ on the z-coordinates), and the TMD ($100 \text{ kJ mol}^{-1} \text{ nm}^{-2}$ on the z-coordinates).

Target configurations for the steered simulations were generated from the most probable dimers identified by the MSM analysis of the TMD-only simulations. Pairwise receptor configurations containing a given dimeric interface in the TMD systems were clustered based on the root mean squared deviation (RMSD) of the participating domains. The least different configurations from all the others for a particular interface was used as a reference to align the protomers of the full-length homology models, so that they could be used as target structures. Steered MD simulations were carried out using the GROMACS 2020.4 software package patched with PLUMED 2.7 (*Tribello et al., 2014*). Starting from the full-length homology models, the systems were steered over 5000 steps with a spring constant of $750 \text{ kJ mol}^{-1} \text{ nm}^{-2}$ for the harmonic potential pulling the system along a collective variable defined by the RMSD of the transmembrane backbone beads to the target structures, using the MOVINGRESTRAINT algorithm in PLUMED. In total, 30 steered simulations were carried out for each interface. Relative free energy estimates were calculated by comparing the work obtained applying Jarzynski's equality (*Jarzynski, 1997*) to the different simulations.

## Calcium imaging

Twenty-four to 48 hr after transfection, cells were imaged at room temperature in extracellular solution composed of 135 mM NaCl, 5.4 mM KCl, 2 mM CaCl$_2$, 1 mM MgCl$_2$, 10 mM HEPES, pH = 7.4. Experiments were conducted on an inverted microscope (Olympus IX73) and imaged with a 20x objective. A continuous gravity-driven perfusion was used throughout the entirety of experiments. GCaMP6f was excited using a 488 nm laser and movies were recorded using a scMOS camera (Hamamatsu ORCA-Flash4v3.0) at 1 Hz with a 100 ms exposure time. Olympus cellSens software was used to select regions of interest (ROIs) (representing single cells or small clusters of 2–3 cells) and subsequent data analysis was performed in Microsoft Excel, in which fluorescence intensities were normalized to a baseline of extracellular solution without drug. Approximately 30 ROIs were analyzed per movie. Glutamate responses were quantified by comparing the response to saturating (1 mM) glutamate. PAM responses were also quantified relative to 1 mM glutamate responses. Concentration-response curves were fit using Graphpad Prism. Multiple independent experiments were performed for each construct tested, including at least two separate transfections.

## Live-cell FRET measurements

Twenty-four to 48 hr after transfection, cells were washed in extracellular solution containing 135 mM NaCl, 5.4 mM KCl, 2 mM CaCl$_2$, 1 mM MgCl$_2$, 10 mM HEPES, pH = 7.4. For inter-LBD FRET experiments, cells were labeled with 1 µM benzylguanine LD555 and 3 µM benzylguanine LD655 (diluted in extracellular solution). For inter-TMD FRET experiments with HaloTag constructs, cells

were labeled with 1 µM JF 549 and 3 µM JF 646 (diluted in extracellular solution). All conditions were labeled for 45 min at 37°C. Following labeling, coverslips were washed in extracellular solution to remove excess dye. Next, coverslips were imaged with a 60x objective on an inverted microscope (Olympus IX73). The donor fluorophore was excited with a 561 nm laser and images were recorded at the same time in both donor and acceptor channels on distinct scMOS cameras (Hamamatsu ORCA-Flash4v3.0) at 0.5–1 Hz with an exposure time of 100 ms. ImageJ was used to analyze cell clusters and FRET was calculated as FRET = $(I_{Acceptor})/(I_{Donor} + I_{Acceptor})$, in which I = fluorescence intensity. Analysis was restricted to relative FRET changes between drug applications rather than absolute FRET values. For each individual trace, FRET was normalized to the basal FRET value observed before drug treatment. For glutamate concentration-response curves, FRET changes were calculated by normalizing to saturating glutamate applied in the same recording. For all concentration-response curves, data was obtained from multiple cell clusters and averaged from at least two separate experiments. All concentration-response curves were fit using Prism (Graphpad). For basal activity measurements, FRET responses to saturating (5 µM) LY34 were normalized to saturating (100 µM) glutamate. A gravity-driven perfusion system was used to deliver drugs diluted in extracellular solution and all experiments were performed at room temperature.

## Patch clamp electrophysiology

Whole cell patch clamp recordings from HEK 293 T cells co-transfected with GIRK channels were performed as previously described (*Gutzeit et al., 2019*).

## Acknowledgements

We thank Melanie Kristt for technical assistance and Drs. Jeremy Dittman, David Eliezer, Olga Boudker and Anant Menon for helpful discussion and Dr. Scott Blanchard for sharing critical reagents. JKT is supported by an NSF Graduate Research Fellowship under grant 1746886. JL is supported by an R35 grant (R35 GM124731) from NIGMS and the Rohr Family Research Scholar Award. Computational work was supported by National Institutes of Health grant DA038882 (to MF and ML). Simulations were run on resources available through the Office of Research Infrastructure of the National Institutes of Health under award numbers S10OD018522 and S10OD026880, as well as the Extreme Science and Engineering Discovery Environment under MCB080077, which is supported by National Science Foundation grant number ACI-1548562.

## Additional information

### Funding

| Funder | Grant reference number | Author |
| --- | --- | --- |
| National Institute of General Medical Sciences | R35 GM124731 | Joshua Levitz |
| National Science Foundation | GRFP | Jordana K Thibado |
| National Institute on Drug Abuse | DA038882 | Marta Filizola Martin J Lohse |
| National Institutes of Health | S10OD018522 | Martin J Lohse |
| National Institutes of Health | S10OD026880 | Martin J Lohse |
| National Science Foundation | MCB080077 | Martin J Lohse |
| National Science Foundation | ACI-1548562 | Martin J Lohse |

The funders had no role in study design, data collection and interpretation, or the decision to submit the work for publication.

### Author contributions

Jordana K Thibado, Conceptualization, Data curation, Formal analysis, Validation, Investigation, Visualization, Writing - original draft, Writing - review and editing; Jean-Yves Tano, Conceptualization,

Resources, Investigation, Methodology, Writing - review and editing; Joon Lee, Conceptualization, Data curation, Formal analysis, Validation, Investigation, Visualization, Methodology; Leslie Salas-Estrada, Conceptualization, Formal analysis, Validation, Investigation, Visualization, Methodology, Writing - review and editing; Davide Provasi, Conceptualization, Formal analysis, Supervision, Validation, Investigation, Methodology, Writing - review and editing; Alexa Strauss, Formal analysis, Investigation, Visualization; Joao Marcelo Lamim Ribeiro, Conceptualization, Formal analysis, Investigation, Visualization, Writing - review and editing; Guoqing Xiang, Formal analysis, Investigation; Johannes Broichhagen, Conceptualization, Resources, Methodology, Writing - review and editing; Marta Filizola, Conceptualization, Data curation, Funding acquisition, Visualization, Methodology, Project administration, Writing - review and editing; Martin J Lohse, Conceptualization, Funding acquisition, Investigation, Methodology, Writing - review and editing; Joshua Levitz, Conceptualization, Data curation, Supervision, Funding acquisition, Validation, Investigation, Visualization, Methodology, Writing - original draft, Project administration, Writing - review and editing

## Author ORCIDs

Jordana K Thibado (ID) https://orcid.org/0000-0001-7293-5364
Davide Provasi (ID) http://orcid.org/0000-0002-2868-303X
Johannes Broichhagen (ID) http://orcid.org/0000-0003-3084-6595
Joshua Levitz (ID) https://orcid.org/0000-0002-8169-6323

## Decision letter and Author response

Decision letter https://doi.org/10.7554/eLife.67027.sa1
Author response https://doi.org/10.7554/eLife.67027.sa2

## Additional files

### Supplementary files

• Transparent reporting form

### Data availability

We have provided source data files for all relevant figures.

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
