## [Decision Letter]

**Acceptance summary:**

In this study, Thibado et al. examine the molecular determinants that govern differential transmembrane dimerization of metabotrobic glutamate receptors mGluR2 and mGluR3. This multifaceted investigation involves sequence analysis, computational modeling, single-molecule pull-down, functional assays and cellular Forster Resonance Energy Transfer, offering rigorous details about the structural factors defining dimerization propensity and the consequential coupling to receptor activity. Particularly interesting is the demonstration that the mGluR2/3 heterodimer displays intermediate dimerization properties between mGluR2 and mGluR3 through TMD4, highlighting mechanisms for fine-tuning activity.

**Decision letter after peer review:**

Thank you for submitting your article "Differences in interactions between transmembrane domains tune the activation of metabotropic glutamate receptors" for consideration by *eLife*. Your article has been reviewed by 3 peer reviewers, one of whom is a member of our Board of Reviewing Editors, and the evaluation has been overseen by Kenton Swartz as the Senior Editor. The following individual involved in review of your submission has agreed to reveal their identity: Terry Hébert (Reviewer #2).

Essential revisions:

In general, the reviewers found this to be an interesting multi-faceted study in which a combination of experiment approaches were used to dissect out the molecular details about transmembrane determinants of mGluR dimerization. While there was enthusiasm for many aspects of this study, some concerns were raised about the structure of the narrative, interpretation of some results and that the concluding model was not well supported.

1. In general, the reviewers found this to be an interesting multi-faceted study in which a combination of experiment approaches were used to dissect out the molecular details about transmembrane determinants of mGluR dimerization. However, there were some concerns that the general narrative of the paper focused on guiding the reader towards the conclusions presented in Figure 8, rather than allow the reader to form their own objective interpretation. Thus, the manuscript should be edited to reduce the proselytizing structure. Furthermore, the model proposed remains hypothetical and is not sufficiently supported by the data presented and so Figure 8 should be removed from the revised manuscript. In addition, the paper needs to be shortened in order to improve readability. This can be achieved by removing sections such as the discussion of the Family A receptor dimerization, and the role of Ala in interfaces, which both do not directly relate to the results in this study.

2. Throughout the paper, the observation of different dimerization proportions has been generally interpreted as a change in stability, but this may not be the case. Since the stoichiometry has been captured from cell membranes, it could be that the dimerization proportion is defined during expression and is kinetically trapped. In this case, the results presented indicate a dimerization signal obtained with certain constructs and expression conditions. On the other hand, "stability" inherently implies an equilibrium affinity, and so use of this description should be reserved for occasions when equilibrium binding has been measured, i.e. when the binding isotherm with titration of the protein density in the membrane, with evidence of reversibility. At the very least, if the dimer signal is examined at a a single density, as is the case in these studies, then it must be shown that the dimerization signal arises from a dynamic equilibrium, e.g. with reversibility or competition with un-labelled species. While the studies presented in Figure 6B,D,F indicate dynamic changes, this is not clear for the data presented elsewhere in the paper.

Therefore, where absolute dimer population is reported (throughout, but especially Figure 7), the wording should be changed to remove terms like "stability", "weaker", "stronger", etc.… Instead, it is appropriate to describe this as an increase/decrease in dimerization signal or propensity. To clarify the interpretation of the dimerization data, please include an explicit description of how the dimerization percentage was calculated in the methods. Specifically address whether this calculation includes a consideration of changes of the expression or does it assume it to be constant? Finally, show the expression data alongside the dimerization data in the main paper, instead of the supplement. This is essential to the interpretation of the results, and is analogous to showing loading control data in western blot figures. Finally, for the expression data, please add error bars and statistical tests.

3. The computational studies are set up in a rigorous manner, with many replicate systems and extended simulation times (the amount of work presented could fill its own paper). However, the connection between the computational analysis and the rest of the paper is a little unclear. For example, many interfaces are identified in the analysis, but the focus remains on TMD4 and TMD6 with the additional interfaces generally ignored. The presentation of the logic between the results from these studies and the rest of the paper should be improved. In addition, the relegation of this work to the supplement has perhaps led to the omission of key details that are important for interpreting the results. Analysis of the sampling achieved, and convergence of results must be included. In the case of the trajectories used for the Markov state analysis, does the protein sample multiple conformations in the same trajectory, or is the sampling of states achieved by the many parallel simulations? A discussion of the lag time analysis was stated, but should be shown. Furthermore, for the free energy estimates from the steered MD, how was convergence over the 30 runs assessed? All of these analyses should be provided even if the data is not presented in the main figures.

*Reviewer #1:*

This is a very interesting study that shows how single-molecule studies of dimerization, coupled with structural and functional analyses, can be used to dissect the mechanisms of mGluR activation. The experiments that are presented here are carefully planned and well executed. Many efforts have been made to properly control the studies, including testing that the complexes captured in the single-molecule pull-down approach reflect the distribution of proteins within the cellular membranes. The paper is overall very strong, but there are a couple of weakness. First, the computational studies that are included here are extensive and set up in a rigorous manner, but there is a lack of data presented to assess the sampling and convergence of results. This information is required for proper interpretation of the data and must be included even when computational studies are presented as supplement. Another concern involves the interpretation of the single-molecule dimerization results. While the dimerization percentage data can stand alone, the interpretation of this data as changes in stability requires evidence that these are dimers that participate in a dynamic equilibrium reaction and that the different constructs are being studied under similar protein densities in the membrane. Notably, the authors have carried out experiments to measure the expression levels by fluorescence microscopy, but the data is not fully quantified with statistical analyses and it should be presented alongside the dimerization results in the main figures. As for evidence of dynamic equilibrium, the data in Figure 6 shows changes in dimerization upon addition of modulators does suggest a dynamic population, but it is not clear that the other dimerization data presented in the paper portrays the equilibrium distribution or a distribution that is established during expression and kinetically trapped instead. Therefore, some test of whether these are truly dynamic populations of dimers should be presented, or care should be taken when discussing these results. Note, many of the conclusions of the study would still hold if these descriptions were changed to "observed dimerization" or "dimer propensity" rather than dimer "stability". With these changes, the study remains to be highly illuminating into the mechanism of mGluR activation and presents many useful approaches for the study of membrane protein oligomerization mechanisms in general.

*Reviewer #2:*

This manuscript begins grounded in previous work from the authors showing that TMDs are involved in mGluR dimerization. Initially they use single molecule pulldowns (SiMPull) to show that receptor TM domains are sufficient to facilitate receptor dimerization (and even oligomerization) in the absence of the large extracellular N-terminal domains. They also identified differences in the relative extent of dimer formation- mGluR2 TMDs were more likely than mGluR3 TMDs to facilitate dimerization alone (i.e. in the absence of a functional LBD-based interface).

Based on structures and homology analysis across the mGluRs they arrive at the idea that TM4, showing the lowest conservation, might be a determinant of the extent of dimerization. Modelling work further suggests that mGluR2 TMD are more likely to dimerize than mGluR3 and also predicted that TM4 might be the difference between the two receptors. Swapping outwardly facing residues between the TM4 segments confirms this supposition experimentally. Functional experiments also showed that there were reciprocal consequences for calcium signalling efficacy mediated by orthosteric or allosteric ligands by swapping TM4 in the two receptors.

Next they showed (as controls) that swapping TM4 did not alter initial LBD movements associated with receptor activation. Next they engineered novel FRET constructs to assess TMD conformational changes. They provide data suggesting the TM4 differences are felt across the receptor in a manner consistent with their previous findings.

They next test the idea that these distinct changes in the two receptors converge on a conserved activation pathway involving TM6- which is identical between the two mGluRs. They provide data showing that TM6 also plays a role in dimerization under basal conditions. Their data is also consistent with the idea that there is a switch from the TM4 to the TM6 interface as part of receptor activation by PAMs or glutamate. They conclude that the TM6 interface reflects the fully activated receptor and the TM4 interface in the inactive receptor- in both mGluRs.

Finally, they turn their attention to heterodimeric forms of the receptor. Using the SiMPull approach again- they show that mGluR2 homodimers are less likely to form than mGluR2/mGluR3 heterodimers but the heterodimer is more likely to form than mGluR3 homodimers. Mutational analysis of the TM4 was consistent with this as well.

They provide a working model for these events which is consistent with their data.

Major strengths – multi-pronged approach to characterizing the dimer interfaces, well-controlled experiments.

Major weaknesses – reliance on a single technique with a chimeric G protein to show functional impacts of TM swaps. Perhaps it would be helpful to compare the stabilities of the LBD-deleted mGluRs with Class A GPCRs. Is the "metastability" of these dimers going to be more similar?

*Reviewer #3:*

Metabotropic glutamate receptors are Family C GPCRs that play important roles in the modulation of neurotransmission in the nervous system. Members of this receptor class operate as dimers, primarily homodimers but also heterodimers, with the orthosteric ligand binding site residing in an N-terminal extracellular region, relatively away from the 7TM region coupling to G proteins. Based on these characteristics, Family C receptors present unique activation mechanisms compared to other GPCR families. Thibado et al. investigated the dimerization properties of the 7TM domains of both mGlu2 and mGlu3 homodimers, and the mGlu2/3 heterodimer in the context of both isolated 7TM domains and full-length receptors using single molecule fluorescence, functional assays, molecular dynamic simulations, and ensemble FRET in cells.

As the authors note, a previous cross-linking study of mGluR2 provided evidence for the close proximity of TM4 and TM5 helices in inactive states and TM6 helices in active states (Xue et al., 2014). In addition, earlier cryoEM studies of Family C receptors have also shown different extents of TM4/5 interfaces in the inactive state and the establishment of a TM6 interface in the active state. Thibado et al. reinforce these findings and build further on the TM4 dimerization concept, starting with sequence conservation analysis and also performing MD simulations. Through mutating/swapping three previously characterized alanine residues on TM4 of mGlu2/3 they show that these sequences determine the propensity of mGlu2/3 homo- and heterodimerization. The results were nicely corroborated with full-length constructs and live cell functional assays. The authors further reveal that these three residues on TM4 affect activation of mGlu2 and mGlu3 both by orthosteric and also allosteric ligands (on the 7TM), although a mechanistic understanding of these effects remains unclear. In a very interesting experimental direction, they also employed new FRET sensors to probe dynamics of TMD dimerization (really nice!). These assays together with mutagenesis revealed differential modulation of the TM4 mediated dimerization of these receptors by orthosteric and allosteric ligands, and further suggested that TM6 is also involved in another dimer interface as previously observed for other family C GPCRs. Based on these results, the authors put forward a rather detailed multistep model of activation involving mGlu2/3 TMD reorientation. Overall, this is a comprehensive study on the 7TM dimerization properties of mGlus, with a focus on mGlu2 and mGlu3, and shows how TM4 residues can control the strength of 7TM interactions and finetune signaling in homodimers and heterodimers. For its most part the work is carefully executed and analyzed with proper quantification and controls.

Specific comments:

The MD simulations show all sorts of interfaces, both symmetric and symmetric with various TM contributors. It is thus difficult to build confidence on these results. I have no doubt that there may be sliding interfaces but there should be more stable configurations if the approach works well with this system, or at least further mutagenesis experimentation to test TM1 or TM3/7 interfaces. Another issue here is that the MDs are run in the presence of only POPC. Could there be a role for cholesterol in these interfaces? For example, the mGlu1 crystal structure revealed a dimer with cholesterol bound at the interface.

The authors attribute the shifts in measured calcium mobilization of the TM4 mutants to the change in dimerization properties of the 7TM. Could there be other contributing factors that have nothing to do with dimerization but affect receptor activation? Even if the mutated residues are facing outwards, TM4 may have a different role related to receptor activation (orthosteric or allosteric) and G protein coupling. I believe it is not straightforward to delineate these effects, and in this regard the proposed explanation comes across as overly simplistic.

Despite a lack of detailed mechanistic insights, the model shown in Figure 8 contains a lot of information that does not appear to be supported by the data. For example, the distance of 7TMs in state i (which oddly shows agonist in the VFTs), the transition between a symmetric 7TM dimer arrangement in state iii to an asymmetric one in state iv, the C-terminal tails and even the coupling to one G protein per dimer which has still not been shown. Do the authors have data to support such a model?

---

## [Author Response]

Essential revisions:In general, the reviewers found this to be an interesting multi-faceted study in which a combination of experiment approaches were used to dissect out the molecular details about transmembrane determinants of mGluR dimerization. While there was enthusiasm for many aspects of this study, some concerns were raised about the structure of the narrative, interpretation of some results and that the concluding model was not well supported.1. In general, the reviewers found this to be an interesting multi-faceted study in which a combination of experiment approaches were used to dissect out the molecular details about transmembrane determinants of mGluR dimerization. However, there were some concerns that the general narrative of the paper focused on guiding the reader towards the conclusions presented in Figure 8, rather than allow the reader to form their own objective interpretation. Thus, the manuscript should be edited to reduce the proselytizing structure. Furthermore, the model proposed remains hypothetical and is not sufficiently supported by the data presented and so Figure 8 should be removed from the revised manuscript. In addition, the paper needs to be shortened in order to improve readability. This can be achieved by removing sections such as the discussion of the Family A receptor dimerization, and the role of Ala in interfaces, which both do not directly relate to the results in this study.

We thank the reviewers for these suggestions. We have revised the writing in the manuscript to allow readers to form objective interpretations and to streamline the text to improve readability as suggested. Please see highlighted changes throughout the manuscript.

While the model presented in Figure 8 was meant merely as a guide to think about how both dynamic TMD dimerization and interface rearrangement may occur during mGluR activation based on both ours and previously published data from other groups, we appreciate the reviewers’ concern that this strayed too far from the data reported. We have thus removed Figure 8 and toned down the discussion to address the implications of our data and place it in the context of the rest of the literature without explicitly linking this to the framework of a specific state model.

2. Throughout the paper, the observation of different dimerization proportions has been generally interpreted as a change in stability, but this may not be the case. Since the stoichiometry has been captured from cell membranes, it could be that the dimerization proportion is defined during expression and is kinetically trapped. In this case, the results presented indicate a dimerization signal obtained with certain constructs and expression conditions. On the other hand, "stability" inherently implies an equilibrium affinity, and so use of this description should be reserved for occasions when equilibrium binding has been measured, i.e. when the binding isotherm with titration of the protein density in the membrane, with evidence of reversibility. At the very least, if the dimer signal is examined at a a single density, as is the case in these studies, then it must be shown that the dimerization signal arises from a dynamic equilibrium, e.g. with reversibility or competition with un-labelled species. While the studies presented in Figure 6B,D,F indicate dynamic changes, this is not clear for the data presented elsewhere in the paper.Therefore, where absolute dimer population is reported (throughout, but especially Figure 7), the wording should be changed to remove terms like "stability", "weaker", "stronger", etc.… Instead, it is appropriate to describe this as an increase/decrease in dimerization signal or propensity.

We thank the reviewers for making this important point and their suggestions for improving the clarity of the text and figures to address this. We have revised the manuscript to remove terms like “stability” and “stronger/weaker” and edited where applicable to “increases/decreases in dimerization propensity” (see highlighted text throughout). In addition, we have added a short section to the discussion (p. 26, paragraph 1) to respond to the aforementioned issues regarding the limitations of SiMPull for addressing true differences in stability versus kinetically trapped populations. We’d like to note that while these limitations warrant future in vitro work to gain a deeper biophysical understanding of mGluR dimerization, the relative advantage of our approach is that it captures receptors directly from the plasma membrane of live cells and is, thus, likely to reflect the cellularly-relevant assembly.

We have also modified Figure 1H to reflect the reviewers’ concerns. We believe that the greater than (“>”) symbol (used in Figure 1H and 7L) is appropriate as it reflects the relative differences in propensity observed rather than implying that an equilibrium measurement has been made as was the case in the initial version. We have clarified this in the figure legend as well. We have also modified Figure 7H to make it clear that we are ranking dimerization *propensity* and not necessarily *affinity*.

To clarify the interpretation of the dimerization data, please include an explicit description of how the dimerization percentage was calculated in the methods. Specifically address whether this calculation includes a consideration of changes of the expression or does it assume it to be constant? Finally, show the expression data alongside the dimerization data in the main paper, instead of the supplement. This is essential to the interpretation of the results, and is analogous to showing loading control data in western blot figures. Finally, for the expression data, please add error bars and statistical tests.

To clarify the interpretation of dimerization data, we have added a description of how the dimerization percentage was calculated to the “Single-molecule pulldown and subunit counting” section of the methods section (page 33). The description is as follows:

“Single molecule fluorescence time traces were manually classified as having 1, 2, 3 or 4 bleaching steps or were discarded if no clear bleaching steps were identified. […] This calculation makes no correction for expression level.

%dimerization=((Observedvalue)−Minvalue)*100Maxvalue−Minvalue”

We have also added expression data alongside the dimerization data in the main and supplemental figures wherever relevant (see Figure 1E, G; Figure 2D, F, H; Figure 2—figure supplement 4C; Figure 2—figure supplement 5C, G; Figure 2—figure supplement 6E; Figure 3—figure supplement 2B, C; Figure 5C; Figure 5—figure supplement 1B, C; Figure 6B, E, H, Figure 7C, F, I, K). All expression plots show individual points, error bars, and statistical tests (unpaired t-tests for 2 conditions or one-way ANOVA plus Tukey-Kramer) are described in the associated figure legends and relevant source files. Note that for 2-color experiments in Figure 7 we now report data on the ratio of expression level of the CLIP-tagged (i.e. prey) and SNAP-tagged (i.e. bait) construct as this is the key parameter for determining the potential effect of expression differences on relative pulldown efficiency. In all cases there are no significant differences in expression ratios between conditions.

3) The computational studies are set up in a rigorous manner, with many replicate systems and extended simulation times (the amount of work presented could fill its own paper). However, the connection between the computational analysis and the rest of the paper is a little unclear. For example, many interfaces are identified in the analysis, but the focus remains on TMD4 and TMD6 with the additional interfaces generally ignored. The presentation of the logic between the results from these studies and the rest of the paper should be improved. In addition, the relegation of this work to the supplement has perhaps led to the omission of key details that are important for interpreting the results. Analysis of the sampling achieved, and convergence of results must be included. In the case of the trajectories used for the Markov state analysis, does the protein sample multiple conformations in the same trajectory, or is the sampling of states achieved by the many parallel simulations? A discussion of the lag time analysis was stated, but should be shown. Furthermore, for the free energy estimates from the steered MD, how was convergence over the 30 runs assessed? All of these analyses should be provided even if the data is not presented in the main figures.

We thank the reviewers for this critique and acknowledgement of the rigorous manner in which the computational study was conducted. Indeed, placing this work in the supplement led to the accidental omission of a few key details, which we have now included in the revised manuscript. We have also moved the description of the CG MD simulations of isolated TMDs into the first section of the results to complement the key initial finding that the mGluR2 TMD shows a higher dimerization propensity than mGluR3. We then revisit the different TMD interfaces in the full-length context in the second part of the Results section to complement our sequence analysis of TM4. We have added the following information:

– A table (Figure 1—supplemental table 1) showing that while most individual trajectories used for the Markov state analysis only sampled one dimeric macrostate, several trajectories sampled more than one dimeric configuration.

– A plot (Figure 1—figure supplement 5) showing the implied timescales as a function of lag time. See accompanying text describing this on page 9, paragraph 1.

– A table (Figure 1—supplemental table 2) showing further information regarding the TM helices involved in the most frequent interfaces associated with each assigned microstate used for the reactive flux analysis.

– Plots (Figure 2—figure supplement 2) illustrating convergence of the free energy differences between representative full-length dimeric configurations of highly populated macrostates.

The contribution of the computational studies stems from their revelation that inter-TMD interactions are highly complex and dynamic with many underlying microstates. However, grouping these microstates into macrostates according to both a >40% helix probability of being at an interface and the largest number of contacts formed by each helix, draws attention to TM1, TM4, and TM5 in the case of mGluR2 or TM3 and TM7 in the case of mGluR3 based on calculated macrostate probabilities larger than 10%. Although these helices are involved in highly probable macrostates in the two receptors, this probability is tuned in the context of a full-length receptor, which favors symmetric TM4 interfaces in mGluR2 over other highly probable ones such as symmetric interfaces involving TM1. We have clarified these observations in the discussion (p. 26, 28), admitting that a complete validation of the computational predictions will require further investigation.